# HOPPER: MULTI-HOP TRANSFORMER FOR SPATIOTEMPORAL REASONING

**Honglu Zhou**[1,*] **Asim Kadav**[2], **Farley Lai**[2], **Alexandru Niculescu-Mizil**[2],
**Martin Renqiang Min**[2], **Mubbasir Kapadia**[1], **Hans Peter Graf**[2]
[1] Department of Computer Science, Rutgers University, Piscataway, NJ, USA
[2] NEC Laboratories America, Inc., San Jose, CA, USA
{hz289,mk1353}@cs.rutgers.edu
{asim,farleylai,alex,renqiang,hpg}@nec-labs.com

## ABSTRACT

This paper considers the problem of spatiotemporal object-centric reasoning in videos. Central to our approach is the notion of *object permanence*, i.e., the ability to reason about the location of objects as they move through the video while being occluded, contained or carried by other objects. Existing deep learning based approaches often suffer from spatiotemporal biases when applied to video reasoning problems. We propose **Hopper**, which uses a Multi-hop Transformer for reasoning object permanence in videos. Given a video and a localization query, **Hopper** reasons over image and object tracks to automatically hop over critical frames in an iterative fashion to predict the final position of the object of interest. We demonstrate the effectiveness of using a contrastive loss to reduce spatiotemporal biases. We evaluate over CATER dataset and find that **Hopper** achieves 73.2% Top-1 accuracy using just 1 FPS by hopping through just a few critical frames. We also demonstrate **Hopper** can perform long-term reasoning by building a CATER-h dataset[1] that requires multi-step reasoning to localize objects of interest correctly.

## 1 INTRODUCTION

In this paper, we address the problem of spatiotemporal object-centric reasoning in videos. Specifically, we focus on the problem of object permanence, which is the ability to represent the existence and the trajectory of hidden moving objects (Baillargeon, 1986). Object permanence can be essential in understanding videos in the domain of: (1) sports like soccer, where one needs to reason, "which player initiated the pass that resulted in a goal?", (2) activities like shopping, one needs to infer "what items the shopper should be billed for?", and (3) driving, to infer "is there a car next to me in the right lane?". Answering these questions requires the ability to detect and understand the motion of objects in the scene. This requires detecting the temporal order of one or more actions of objects. Furthermore, it also requires learning *object permanence*, since it requires the ability to predict the location of non-visible objects as

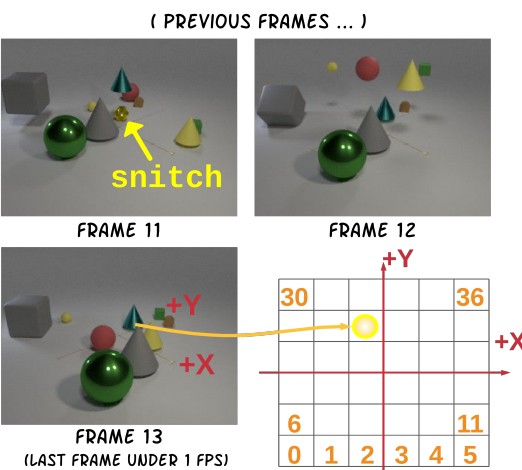

**Figure 1:** Snitch Localization in CATER (Girdhar & Ramanan, 2020) is an object permanence task where the goal is to classify the final location of the snitch object within a 2D grid space.

---

*Work done as a NEC Labs intern.

[1]https://github.com/necla-ml/cater-h

they are occluded, contained or carried by other objects (Shamsian et al., 2020). Hence, solving this task requires compositional, multi-step spatiotemporal reasoning which has been difficult to achieve using existing deep learning models (Bottou, 2014; Lake et al., 2017).

Existing models have been found lacking when applying to video reasoning and object permanence tasks (Girdhar & Ramanan, 2020). Despite rapid progress in video understanding benchmarks such as action recognition over large datasets, deep learning based models often suffer from spatial and temporal biases and are often easily fooled by statistical spurious patterns and undesirable dataset biases (Johnson et al., 2017b). For example, researchers have found that models can recognize the action "swimming" even when the actor is masked out, because the models rely on the swimming pool, the scene bias, instead of the dynamics of the actor (Choi et al., 2019).

Hence, we propose **Hopper** to address debiased video reasoning. **Hopper** uses multi-hop reasoning over videos to reason about object permanence. Humans realize object permanence by identifying *key* frames where objects become hidden (Bremner et al., 2015) and reason to predict the motion and final location of objects in the video. Given a video and a localization query, **Hopper** uses a Multi-hop Transformer (MHT) over image and object tracks to automatically identify and hop over *critical* frames in an iterative fashion to predict the final position of the object of interest. Additionally, **Hopper** uses a contrastive debiasing loss that enforces consistency between attended objects and correct predictions. This improves model robustness and generalization. We also build a new dataset, CATER-h, that reduces temporal bias in CATER and requires long-term reasoning.

We demonstrate the effectiveness of **Hopper** over the recently proposed CATER 'Snitch Localization' task (Girdhar & Ramanan, 2020) (Figure 1). **Hopper** achieves 73.2% Top-1 accuracy in this task at just 1 FPS. More importantly, **Hopper** identifies the critical frames where objects become invisible or reappears, providing an interpretable summary of the reasoning performed by the model. To summarize, the contributions of our paper are as follows: First, we introduce **Hopper** that provides a framework for multi-step compositional reasoning in videos and achieves state-of-the-art accuracy in CATER object permanence task. Second, we describe how to perform interpretable reasoning in videos by using iterative reasoning over critical frames. Third, we perform extensive studies to understand the effectiveness of multi-step reasoning and debiasing methods that are used by **Hopper**. Based on our results, we also propose a new dataset, CATER-h, that requires longer reasoning hops, and demonstrates the gaps of existing deep learning models.

## 2  RELATED WORK

**Video understanding.**  Video tasks have matured quickly in recent years (Hara et al., 2018); approaches have been migrated from 2D or 3D ConvNets (Ji et al., 2012) to two-stream networks (Simonyan & Zisserman, 2014), inflated design (Carreira & Zisserman, 2017), models with additional emphasis on capturing the temporal structures (Zhou et al., 2018), and recently models that better capture spatiotemporal interactions (Wang et al., 2018; Girdhar et al., 2019). Despite the progress, these models often suffer undesirable dataset biases, easily confused by backgrounds objects in new environments as well as varying temporal scales (Choi et al., 2019). Furthermore, they are unable to capture reasoning-based constructs such as causal relationships (Fire & Zhu, 2017) or long-term video understanding (Girdhar & Ramanan, 2020).

**Visual and video reasoning.**  Visual and video reasoning have been well-studied recently, but existing research has largely focused on the task of question answering (Johnson et al., 2017a; Hudson & Manning, 2018; 2019a; Yi et al., 2020). CATER, a recently proposed diagnostic *video recognition* dataset focuses on spatial and temporal reasoning as well as localizing particular object of interest. There also has been significant research in object tracking, often with an emphasis on occlusions with the goal of providing object permanence (Wojke et al., 2017; Wang et al., 2019b). Traditional object tracking approaches often require expensive supervision of location of the objects in every frame. In contrast, we address object permanence and video recognition on CATER with a model that performs tracking-integrated object-centric reasoning without this strong supervision.

**Multi-hop reasoning.** Reasoning systems vary in expressive power and predictive abilities, which include symbolic reasoning, probabilistic reasoning, causal reasoning, etc. (Bottou, 2014). Among them, multi-hop reasoning is the ability to reason with information collected from multiple passages to derive the answer (Wang et al., 2019a), and it gives a discrete intermediate output of the reasoning process, which can help gauge model's behavior beyond just the final task accuracy (Chen et al., 2019). Several multi-hop datasets and models have been proposed for the reading comprehension

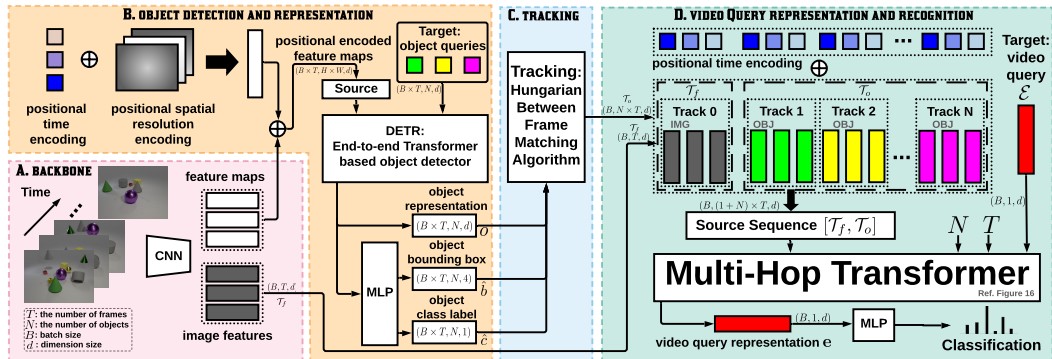

**Figure 2:** An overview of the `Hopper` framework. `Hopper` first obtains frame representations from the input video. Object representations and object tracks are then computed to enable *tracking-integrated object-centric reasoning* for the Multi-hop Transformer (details in Section 4).

task (Welbl et al., 2018; Yang et al., 2018b; Dua et al., 2019; Dhingra et al., 2020). We extend multi-hop reasoning to the video domain by developing a dataset that explicitly requires aggregating clues from different spatiotemporal parts of the video, as well as a multi-hop model that automatically extracts a step-by-step reasoning chain, which improves interpretability and imitates a natural way of thinking. We provide an extended discussion of related work in Appendix I.

## 3   HOPPER

`Hopper` (Figure 2) is a framework inspired from the observation that humans think in terms of entities and relations. Unlike traditional deep visual networks that perform processing over the pixels from which they learn and extract features, object-centric learning-based architecture explicitly separates information about entities through grouping and abstraction from the low-level information (Locatello et al., 2020). `Hopper` obtains representations of object entities from the low-level pixel information of every frame (Section 3.2). Additionally, to maintain object permanence, humans are able to identify key moments when the objects disappear and reappear. To imitate that, `Hopper` computes object tracks with the goal to have a more consistent object representation (Section 3.3) and then achieves *multi-step compositional long-term reasoning* with the Multi-hop Transformer to pinpoint these critical moments. Furthermore, `Hopper` combines both fine-grained (object) and coarse-grained (image) information to form a contextual understanding of a video. As shown in Figure 2, `Hopper` contains 4 components; we describe them below.

### 3.1   BACKBONE

Starting from the initial RGB-based video representation $x_{\mathrm{v}} \in \mathbb{R}^{T \times 3 \times H_0 \times W_0}$ where $T$ represents the number of frames of the video, 3 is for the three color channels, and $H_0$ and $W_0$ denote the original resolution height and width, a conventional CNN backbone would extract the feature map $f \in \mathbb{R}^{T \times P \times H \times W}$ and for every frame $t$ a compact image representation $i_t \in \mathbb{R}^P$. The backbone we use is ResNeXt-101 from Ma et al. (2018), $P = 2048$ and $H, W = 8, 10$. A $1 \times 1$ convolution (Carion et al., 2020) then reduces the channel dimension of $f$ from $P$ to a smaller dimension $d$ ($d = 256$), and a linear layer is used to turn the dimension of $i_t$ from $P$ to $d$.

### 3.2   OBJECT DETECTION AND REPRESENTATION

We collapse the spatial dimensions into 1 dimension and combine the batch dimension with the temporal dimension for the feature map $f$. Positional encodings are learned for each time step ($T$ in total) and each spatial location ($H \times W$ in total), which are further added to the feature map in an element-wise manner. The positional encoding-augmented feature map is the source input to the transformer encoder (Vaswani et al., 2017) of DETR (Carion et al., 2020). DETR is a recently proposed transformer-based object detector for image input; it additionally accepts $N$ embeddings of object queries for every image (assuming every image at most has $N$ objects[2]) to the transformer decoder. We also combine the batch dimension with temporal dimension for the object queries. Outputs from DETR are transformed object representations that are used as inputs to a multilayer perceptron (MLP) to predict the bounding box and class label of every object. For Snitch Localization, DETR is trained on object annotations from LA-CATER (Shamsian et al., 2020).

---

[2]$\varnothing$, i.e., none object, will be predicted if the number of objects in an image is less than $N$.

### 3.3 TRACKING

Tracking produces consistent object representations as it links the representations of each object through time. We perform tracking using the unordered object representations, bounding boxes and labels as inputs, and applying our Hungarian-based algorithm to match objects between every two consecutive frames. We describe the details as follows.

Tracking is essentially an association problem (Bewley et al., 2016). An association between 2 objects respectively from consecutive 2 frames can be defined by the object class agreement and the difference of the two bounding boxes. Let us denote by $\hat{y} = [\hat{y}_t]_{t=1}^T$ the predicted list of objects at all frames in a video, where $\hat{y}_t = \{\hat{y}_t^i\}_{i=1}^N$ denotes the predicted set of objects at frame $t$. Each object is represented as a 4-tuple $\hat{y}_t^i = (\hat{c}_t^i, \hat{b}_t^i, \{\hat{p}_t^i(c) | c \in C\}, o_t^i)$ where $\hat{c}_t^i$ denotes the class label that has the maximum predicted likelihood for object $i$ at frame $t$, $\hat{b}_t^i \in [0, 1]^4$ is a vector that defines the bounding box top left and bottom right coordinates relative to the image size, $\hat{p}_t^i(c)$ denotes the predicted likelihood for class $c$ (where $C = \{$large metal green cube, small metal green cube, . . . , $\varnothing\}$), and $o_t^i \in \mathbb{R}^d$ denotes the representation vector of this object $i$ at frame $t$.

In order to obtain the optimal bipartite matching between the set of predicted objects at frame $t$ and $t + 1$, we search for a permutation of $N$ elements $\sigma \in \mathfrak{S}_N$ with the lowest permutation cost:

$$\hat{\sigma} = \underset{\sigma \in \mathfrak{S}_N}{\arg \min} \sum_{i=1}^N \mathscr{L}_{\text{track}} \left( \hat{y}_t^i, \hat{y}_{t+1}^{\sigma(i)} \right) \tag{1}$$

where $\mathscr{L}_{\text{track}}$ is a pair-wise track matching cost between predicted object $\hat{y}_t^i$ (i.e., object $i$ at frame $t$) and predicted object at frame $t + 1$ with index $\sigma(i)$ from the permutation $\sigma$, denoted by $\hat{y}_{t+1}^{\sigma(i)}$. Following Carion et al. (2020), the optimal assignment is computed efficiently with the Hungarian algorithm. The track matching cost at time $t$ for object $i$ is defined as

$$\mathscr{L}_{\text{track}} \left( \hat{y}_t^i, \hat{y}_{t+1}^{\sigma(i)} \right) = -\lambda_{\text{c}} \mathbb{1}_{\{\hat{c}_t^i \neq \varnothing\}} \hat{p}_{t+1}^{\sigma(i)} \left( \hat{c}_t^i \right) + \lambda_{\text{b}} \mathbb{1}_{\{\hat{c}_t^i \neq \varnothing\}} \mathscr{L}_{\text{box}} \left( \hat{b}_t^i, \hat{b}_{t+1}^{\sigma(i)} \right) \tag{2}$$

where $\mathbb{1}$ denotes an indicator function such that the equation after the symbol $\mathbb{1}$ only takes effect when the condition inside the $\{\dots\}$ is true, otherwise the term will be 0. $\lambda_{\text{c}}, \lambda_{\text{b}} \in \mathbb{R}$ weight each term. $\mathscr{L}_{\text{box}}$ is defined as a linear combination of the $L_1$ loss and the generalized IoU loss (Rezatofighi et al., 2019). When the predicted class label of object $i$ at frame $t$ is not $\varnothing$, we aim to maximize the likelihood of the class label $\hat{c}_t^i$ for the predicted object $\sigma(i)$ at frame $t + 1$, and minimize the bounding box difference between the two. The total track matching cost of a video is the aggregation of $\mathscr{L}_{\text{track}} \left( \hat{y}_t^i, \hat{y}_{t+1}^{\sigma(i)} \right)$ from object $i = 1$ to $N$ and frame $t = 1$ to $T - 1$.

This Hungarian-based tracking algorithm is used due to its simplicity. A more sophisticated tracking solution (e.g. DeepSORT (Wojke et al., 2017)) could be easily integrated into **Hopper**, and may improve the accuracy of tracking in complex scenes.

### 3.4 VIDEO QUERY REPRESENTATION AND RECOGNITION

The $N$ object tracks obtained from the Hungarian algorithm and a single track of image features from the backbone are further added with the learned positional time encodings to form the source input to our Multi-hop Transformer (which will be introduced in Section 4). Multi-hop Transformer produces the final latent representation of the video query $\mathbf{e} \in \mathbb{R}^d$. A MLP uses the video query representation $\mathbf{e}$ as an input and predicts the grid class for the Snitch Localization task.

## 4 MULTI-HOP TRANSFORMER

Motivated by how humans reason an object permanence task through identifying *critical moments of key objects* in the video (Bremner et al., 2015), we propose Multi-hop Transformer (MHT). MHT reasons by hopping over frames and selectively attending to objects in the frames, until it arrives at the correct object that is the most important for the task. MHT operates in an iterative fashion, and each iteration produces one-hop reasoning by selectively attending to objects from a collection of frames. Objects in that collection of frames form the candidate pool of that hop. Later iterations are built upon the knowledge collected from the previous iterations, and the size of the candidate pool decreases as iteration runs. We illustrate the MHT in Figure 16 in Appendix. The overall module is described in Algorithm 1. MHT accepts a frame track $\mathcal{T}_f$: $[i_1, i_2, \cdots, i_T]$, an object track $\mathcal{T}_o$: $[o_1^1, o_2^1, \cdots, o_T^1, \cdots, o_1^N, o_2^N, \cdots, o_T^N]$, an initial target video query embedding $\mathcal{E}$, the number of objects

$N$ and number of frames $T$. $h$ denotes the hop index, and $t$ is the frame index that the previous hop (i.e., iteration) mostly attended to, in Algorithm 1.

**Overview.** Multiple iterations are applied over MHT, and each iteration performs one hop of reasoning by attending to certain objects in critical frames. With a total of $H$ hops, MHT produces refined representation of the video query $\mathcal{E}$ ($\mathcal{E} \in \mathbb{R}^{1 \times d}$). As the complexity of video varies, $H$ should also vary across videos. In addition, MHT operates in an autoregressive manner to process the incoming frames. This is achieved by 'Masking()' (will be described later). The autoregressive processing in MHT allows hop $h+1$ to only attend to objects in frames after frame $t$, if hop $h$ mostly attends to an object at frame $t$. We define *the most attended object of a hop* as the object that has the highest attention weight (averaged from all heads) from the encoder-decoder multi-head attention layer in Transformer$_s$ (will be described later). The hopping ends when the most attended object is an object in the last frame.

**MHT Architecture.** Inside of the MHT, there are 2 encoder-decoder transformer units (Vaswani et al., 2017): Transformer$_f$ and Transformer$_s$. This architecture is inspired by study in cognitive science that reasoning consists of 2 stages: first, one has to establish the domain about which one reasons and its properties, and only after this initial step can one's reasoning happen (Stenning & Van Lambalgen, 2012). We first use Transformer$_f$ to adapt the representations of the object entities, which form the main ingredients of the domain under the context of an object-centric video task. Then, Transformer$_s$ is used to produce the task-oriented representation to perform reasoning.

We separate 2 types of information from the input: *attention candidates* and *helper information*. This separation comes from the intuition that humans sometimes rely on additional information outside of the candidate answers. We call such *additional information* as helper information $\mathcal{H}$ (specifically in our case, could be the coarse-grained global image context, or information related to the previous reasoning step). We define *candidate answers* as attention candidates $\mathcal{U}$, which are representations of object entities (because object permanence is a task that requires reasoning relations of objects). For each hop, we first extract *attention candidates* and *helper information* from the source sequence, then use Transformer$_f$ to condense the most useful information by attending to attention candidates via self-attention and helper information via encoder-decoder attention. After that, we use Transformer$_s$ to learn the latent representation of the video query by attentively utilizing the information extracted from Transformer$_f$ (via encoder-decoder attention). Thus, MHT decides on which object to mostly attend to, given the *current* representation of the video query $\mathcal{E}$, by reasoning about the relations between the object entities ($\mathcal{U}$), and how would each object entity relate to the reasoning performed by the previous hop *or* global information ($\mathcal{H}$).

---

**Algorithm 1** Multi-hop Transformer module.

---

**Input**: $\mathcal{T}_f \in \mathbb{R}^{T \times d}$, $\mathcal{T}_o \in \mathbb{R}^{NT \times d}$, $\mathcal{E} \in \mathbb{R}^{1 \times d}$, $N \in \mathbb{R}$, $T \in \mathbb{R}$
**Params**: LayerNorm, Transformer$_f$, Transformer$_s$, $W_g$, $b_g$

1: $h \leftarrow 0, t \leftarrow 0$
2: **while** $t \neq (T-1)$ **do**
3: $\quad h \leftarrow h + 1$
4: $\quad$ **if** $h > 1$ **then**
5: $\quad\quad \mathcal{H} \leftarrow$ Extract $(\mathcal{T}_o, N, T, t)$
6: $\quad$ **else**
7: $\quad\quad \mathcal{H} \leftarrow \mathcal{T}_f$
8: $\quad$ **end if**
9: $\quad \mathcal{U} \leftarrow \mathcal{T}_o$
10: $\quad \mathcal{U}_{\text{update}}, \_ \leftarrow$ Transformer$_f$ $(\mathcal{U}, \mathcal{H})$
11: $\quad \mathcal{U}_{\text{update}} \leftarrow$ Sigmoid $(W_g \cdot \mathcal{U}_{\text{update}} + b_g) \odot \mathcal{U}$
12: $\quad \mathcal{U}_{\text{mask}} \leftarrow$ Masking $(\mathcal{U}_{\text{update}}, t)$
13: $\quad \mathcal{E}, \mathcal{A} \leftarrow$ Transformer$_s$ $(\mathcal{U}_{\text{mask}}, \mathcal{E})$
14: $\quad t \leftarrow$ Softargmax $(\mathcal{A})$
15: **end while**
16: $\mathbf{e} \leftarrow$ LayerNorm $(\mathcal{E})$
**Return e**

---

**Transformer$_f$.** Transformer$_f$ uses helper information $\mathcal{H}$ from the previous hop, to adapt the representations of the object entities $\mathcal{U}$ to use in the current reasoning step. Formally, $\mathcal{U}$ is the object track sequence $\mathcal{T}_o$ as in line 9 in Algorithm 1 ($\mathcal{U} \in \mathbb{R}^{NT \times d}$, $NT$ tokens), whereas $\mathcal{H}$ encompasses different meanings for hop 1 and the rest of the hops. For hop 1, $\mathcal{H}$ is the frame track $\mathcal{T}_f$ ($\mathcal{H} \in \mathbb{R}^{T \times d}$, $T$ tokens, line 7). This is because hop 1 is necessary for all videos with the goal to find the first critical object (and frame) from *the global information*. Incorporating frame representations is also beneficial because it provides complementary information and can mitigate occasional errors from the object detector and tracker. For the rest of the hops, $\mathcal{H}$ is the set of representations of all objects in the frame that *the previous hop mostly attended to* ($\mathcal{H} \in \mathbb{R}^{N \times d}$, $N$ tokens, Extract() in line 5). The idea is that, to select an answer from object candidates after frame $t$, objects in frame $t$ could be the most important helper information. Transformer$_f$ produces $\mathcal{U}_{\text{update}}$

($\mathcal{U}_{\text{update}} \in \mathbb{R}^{NT \times d}$, $NT$ tokens), an updated version of $\mathcal{U}$, by selectively attending to $\mathcal{H}$. Further, MHT conditionally integrates helper-fused representations and the original representations of $\mathcal{U}$. This conditional integration is achieved by *Attentional Feature-based Gating* (line 11), with the role to combine the new modified representation with the original representation. This layer, added on top of Transformer$_f$, provides additional new information, because it switches the perspective into learning new representations of object entities by learning a feature mask (values between 0 and 1) to select salient dimensions of $\mathcal{U}$, conditioned on the adapted representations of object entities that are produced by Transformer$_f$. Please see details about this layer in Margatina et al. (2019)).

**Transformer$_s$.** Transformer$_s$ is then used to produce the task-oriented video query representation $\mathcal{E}$. As aforementioned, MHT operates in an autoregressive manner to proceed with time. This is achieved by 'Masking()' that turns $\mathcal{U}_{\text{update}}$ into $\mathcal{U}_{\text{mask}}$ ($\mathcal{U}_{\text{mask}} \in \mathbb{R}^{NT \times d}$) for Transformer$_s$ by only retaining the object entities in frames *after* the frame that the previous hop mostly attended to (for hop 1, $\mathcal{U}_{\text{mask}}$ is $\mathcal{U}_{\text{update}}$). Masking is commonly used in NLP for the purpose of autoregressive processing (Vaswani et al., 2017). Masked objects will have 0 attention weights. Transformer$_s$ learns representation of the video query $\mathcal{E}$ by attending to $\mathcal{U}_{\text{mask}}$ (line 13). It indicates that, unlike Transformer$_f$ in which message passing is performed across all connections between tokens in $\mathcal{U}$, between tokens in $\mathcal{H}$, and especially across $\mathcal{U}$ and $\mathcal{H}$ (we use $\mathcal{U}$ for Transformer$_f$, instead of $\mathcal{U}_{\text{mask}}$, because potentially, to determine which object a model should mostly attend to in frames after $t$, objects in and before frame $t$ might also be beneficial), message passing in Transformer$_s$ is only performed between tokens in $\mathcal{E}$ (which has only 1 token for Snitch Localization), between tokens in *unmasked* tokens in $\mathcal{U}_{\text{update}}$, and more importantly, across connections between the video query $\mathcal{E}$ and *unmasked* tokens in $\mathcal{U}_{\text{update}}$. The indices of the most attended object and the frame that object is in, are determined by attention weights $\mathcal{A}$ from the previous hop with a *differentiable* 'Softargmax()' (Chapelle & Wu, 2010; Honari et al., 2018), defined as, $\text{softargmax}(x) = \sum_i \frac{e^{\beta x_i}}{\sum_j e^{\beta x_j}} i$, where $\beta$ is an arbitrarily large number. Attention weights $\mathcal{A}$ ($\mathcal{A} \in \mathbb{R}^{NT \times 1}$) is averaged from all heads. $\mathcal{E}$ is updated over the hops, serving the information exchange between the hops.

**Summary & discussion.** $\mathcal{H}$, $\mathcal{U}_{\text{mask}}$ and $\mathcal{E}$ are updated in every hop. $\mathcal{E}$ should be seen as an encoding for a query of the entire video. Even though in this dataset, a single token is used for the video query, and the self-attention in the decoder part of Transformer$_s$ is thus reduced to a stacking of 2 linear transformations, it is possible that multiple queries would be desirable in other applications. These structural priors that are embedded in (e.g., the iterative hopping mechanism and attention, which could be treated as a soft tree) essentially provide the composition rules that algebraically manipulate the previously acquired knowledge and lead to the higher forms of reasoning (Bottou, 2014). Moreover, MHT could potentially correct errors made by object detector and tracker, but poor performance of them (especially object detector) would also make MHT suffer because (1) inaccurate object representations will confuse MHT in learning, and (2) the heuristic-based loss for intermediate hops (will be described in Section 5) will be less accurate.

## 5 TRAINING

We propose the following training methods for the Snitch Localization task and present an ablation study in Appendix A. We provide the implementation details of our model in Appendix H.

▶ *Dynamic hop stride.* A basic version of autoregressive MHT is to set the per-hop frame stride to 1 with 'Masking()' as usually done in NLP. It means that Transformer$_s$ will only take in objects *in* frame $t$+1 as the source input if the previous hop mostly attended to an object in frame $t$. However, this could produce an unnecessary long reasoning chain. By using dynamic hop stride, we let the model automatically decide on which upcoming frame to reason by setting 'Masking()' to give unmasked candidates as objects in *frames* after the frame that the previous hop mostly attended to.

▶ *Minimal hops of reasoning.* We empirically set the minimal number of hops that the model has to perform for any video as 5 to encourage multi-hop reasoning with reasonably large number of hops (unless not possible, e.g., if the last visible snitch is in the second last frame, then the model is only required to do 2 hops). This is also achieved by 'Masking()'. E.g., if hop 1 mostly attends to an object in frame 3, 'Masking()' will *not* mask objects in frames from frame 4 to frame 10 for hop 2, in order to allow hop 3, 4, 5 to happen (suppose 13 frames per video, and frame 4 is computed from $3 + 1$, frame 10 is computed as $max(3 + 1, 13 - (5 - 2))$).

▶ *Auxiliary hop 1 object loss.* Identifying the correct object to attend to in early hops is critical and for Snitch Localization, the object to attend to in hop 1 should be the last visible snitch (intuitively).

Hence, we define an auxiliary hop 1 object loss as the cross-entropy of classifying index of the last visible snitch. Inputs to this loss are the computed index of the last visible snitch from $\mathcal{T}_o$ (with the heuristic that approximates it from predicted object bounding boxes and labels), as well as the attention weights $\mathcal{A}$ from Transformer$_s$ of hop 1, serving as predicted likelihood for each index.

▶ *Auxiliary hop* 2 *object loss.* Similarly, we let the second hop to attend to the immediate occluder or container of the last visible snitch. The auxiliary hop 2 object loss is defined as the cross-entropy of classifying index of the immediate occluder or container of the last visible snitch. Inputs to this loss are the heuristic [3] computed index and attention weights $\mathcal{A}$ from Transformer$_s$ of hop 2.

▶ *Auxiliary hop* 1&2 *frame loss.* Attending to objects in the *correct frames* in hop 1 and 2 is critical for the later hops. A $L_1$ loss term could guide the model to find out the correct frame index.

▶ *Teacher forcing* is often used as a strategy for *training* recurrent neural networks that uses the ground truth from a prior time step as an input (Williams & Zipser, 1989). We use teacher forcing for hop 2 and 3 by providing the ground truth $\mathcal{H}$ and $\mathcal{U}_{\mathrm{mask}}$ (since we can compute the frame index of the last visible snitch with heuristics as described above).

▶ *Contrastive debias loss via masking out.* This loss is inspired from the *human mask confusion loss* in Choi et al. (2019). It allows penalty for the model if it could make predictions correctly when the most attended object in the last frame is masked out. However, in contrast to human mask, we enforce consistency between attended objects and correct predictions, ensuring that the model *understands* why it is making a correct prediction. The idea here is that the model should not be able to predict the correct location without seeing the correct evidence. Technically, the contrastive debias loss is defined as the entropy function that we hope to maximize, defined as follows.

$$\mathscr{L}_{\mathrm{debias}} = \mathbb{E}\left[\sum_{k=1}^{K} g_\theta\left(\mathcal{M}_{\mathrm{neg}}; \cdots\right)\left(\log g_\theta\left(\mathcal{M}_{\mathrm{neg}}; \cdots\right)\right)\right] \tag{3}$$

where $g_\theta$ denotes the video query representation and recognition module (Multi-hop Transformer along with MLP) with parameter $\theta$ that produces the likelihood of each grid class, $\mathcal{M}_{\mathrm{neg}}$ is the source sequence to the Multi-hop Transformer with the most attended object in the last hop being masked out (set to zeros), and $K$ denotes the number of grid classes.

**Summary & discussion.** The total loss of the model is a linear combination of hop 1 and hop 2 object & frame loss, contrastive debiasing loss for the last hop, and the final grid classification cross-entropy loss. The object & frame loss for hop 1 and 2 are based on heuristics. The motivation is to provide weak supervision for the early hops to avoid error propagation, as multi-hop model can be difficult to train, without intermediate supervision or when ground truth reasoning chain is not present (as in Hopper) (Dua et al., 2020; Qi et al., 2019; Ding et al., 2019; Wang et al., 2019a; Chen & Durrett, 2018; Jiang & Bansal, 2019). One can use similar ideas as the ones here on other tasks that require multi-hop reasoning (e.g., design self-supervision or task-specific heuristic-based weak supervision for intermediate hops, as existing literature often does).

## 6 EXPERIMENTS

**Datasets.** Snitch Localization (Figure 1) is the most challenging task in the CATER dataset and it requires maintaining object permanence to solve the task successfully (Girdhar & Ramanan, 2020). However, CATER is highly imbalanced for Snitch Localization task in terms of the temporal cues: snitch is entirely visible at the end of the video for $58\%$ samples, and entirely visible at the second last frame for $14\%$ samples. As a result, it creates a temporal bias in models to predict based on the last few frames. To address temporal bias in CATER, we create a new dataset, CATER-hard (CATER-h for short), with diverse temporal variations. In CATER-h, every frame index roughly shares an equal number of videos, to have the last visible snitch in that frame (Figure 13).

**Baselines & metrics.** We experiment with TSM (Lin et al., 2019), TPN (Yang et al., 2020), and SINet (Ma et al., 2018). Additionally, Transformer that uses the time-encoded frame track as the source sequence is utilized. Both SINet and Transformer are used to *substitute* our novel MHT in the **Hopper** framework in order to demonstrate the effectiveness of the proposed MHT. This means that **Hopper**-transformer and **Hopper**-sinet use the same representations of image track and object tracks as our **Hopper**-multihop. Moreover, we report results from a Random baseline, a Tracking baseline used by Girdhar & Ramanan (2020) (DaSiamRPN), and a Tracking baseline based on our Hungarian algorithm (Section 3.3) in order to have a thorough understanding of how

---

[3]The heuristic is $L_1$ distance to find out that in the immediate frame which object's bounding box bottom midpoint location is closest to that of the last visible snitch.

well our tracking component performs. For CATER. we also compare with the results in Girdhar & Ramanan (2020). Details of the baselines are available in Appendix H. We evaluate models using Top-1 and Top-5 accuracy, as well as mean $L_1$ distance of the predicted grid cell from the ground truth following Girdhar & Ramanan (2020). $L_1$ is cognizant of the grid structure and will penalize confusion between adjacent cells less than those between distant cells.

| Methods | FPS | # Frames | Top 1 $\uparrow$ | Top 5 $\uparrow$ | $L_1 \downarrow$ |
|---|---|---|---|---|---|
| Random | - | - | 2.8 | 13.8 | 3.9 |
| DaSiamRPN (Tracking) (Zhu et al., 2018) | - | - | 33.9 | 40.8 | 2.4 |
| Hungarian (Tracking - ours) | - | - | 46.0 | 52.7 | 1.9 |
| TSN (RGB) (Wang et al., 2016) | - | 3 | 14.1 | 38.5 | 3.2 |
| TSN (RGB) + LSTM (Wang et al., 2016) | - | 3 | 25.6 | 67.2 | 2.6 |
| TSN (Flow) (Wang et al., 2016) | - | 3 | 9.6 | 32.3 | 3.7 |
| TSN (Flow) + LSTM (Wang et al., 2016) | - | 3 | 14.0 | 43.5 | 3.2 |
| I3D-50 (Carreira & Zisserman, 2017) | 5 | 64 | 57.4 | 78.4 | 1.4 |
| I3D-50 + LSTM (Carreira & Zisserman, 2017) | 5 | 64 | 60.2 | 81.8 | 1.2 |
| I3D-50 + NL (Wang et al., 2018) | 2.5 | 32 | 26.7 | 68.9 | 2.6 |
| I3D-50 + NL + LSTM (Wang et al., 2018) | 2.5 | 32 | 46.2 | 69.9 | 1.5 |
| TPN-101 (Yang et al., 2020) | 2.5 | 32 | *65.3** | 83.0 | 1.09 |
| TSM-50 (Lin et al., 2019) | 1 | 13 | 64.0 | 85.7 | ***0.93*** |
| SINet (Ma et al., 2018) | 1 | 13 | 21.1 | 47.1 | 3.14 |
| Transformer (Vaswani et al., 2017) | 1 | 13 | 13.7 | 39.9 | 3.53 |
| **Hopper**-transformer (last frame) | 1 | 13 | 61.1 | 86.6 | 1.42 |
| **Hopper**-transformer | 1 | 13 | 64.9 | *90.1** | 1.11 |
| **Hopper**-sinet | 1 | 13 | ***69.1*** | ***91.8*** | *1.02** |
| **Hopper**-multihop (our proposed method) | 1 | 13 | ***73.2⋆*** | ***93.8⋆*** | ***0.85⋆*** |

**Table 1:** CATER Snitch Localization results (on the test set). The top 3 performance scores are highlighted as: *First*⋆, **Second**, *Third*\*. **Hopper** outperforms existing methods under only 1 FPS.

| Methods | FPS | # Frames | Top 1 $\uparrow$ | Top 5 $\uparrow$ | $L_1 \downarrow$ |
|---|---|---|---|---|---|
| Random | - | - | 2.4 | 13.6 | 3.9 |
| DaSiamRPN (Tracking) (Zhu et al., 2018) | - | - | 17.1 | 26.1 | 2.9 |
| Hungarian (Tracking - ours) | - | - | 37.2 | 44.4 | 2.3 |
| TPN-101 (Yang et al., 2020) | 2.5 | 32 | 50.2 | *88.3** | 1.46 |
| TSM-50 (Lin et al., 2019) | 1 | 13 | 44.0 | 75.7 | 1.54 |
| SINet (Ma et al., 2018) | 1 | 13 | 18.6 | 44.3 | 3.24 |
| Transformer (Vaswani et al., 2017) | 1 | 13 | 11.6 | 34.4 | 3.49 |
| **Hopper**-transformer (last frame) | 1 | 13 | 41.8 | 79.3 | 2.10 |
| **Hopper**-transformer | 1 | 13 | *57.6** | ***90.1*** | *1.39** |
| **Hopper**-sinet | 1 | 13 | ***62.8*** | *91.7** | ***1.25*** |
| **Hopper**-multihop (our proposed method) | 1 | 13 | ***68.4⋆*** | 87.9 | ***1.09⋆*** |

**Table 2:** CATER-h Snitch Localization results (on the test set). The top 3 performance scores are highlighted as: *First*⋆, **Second**, *Third*\*. **Hopper** outperforms existing methods under only 1 FPS.

**Results.** We present the results on CATER in Table 1 and on CATER-h in Table 2. For all methods, we find that the performance on CATER-h is lower than that on CATER, which demonstrates the difficulty of CATER-h. Such performance loss is particularly severe for the tracking baseline and the temporal video understanding methods (TPN and TSM). The DaSiamRPN tracking approach only solves about a third of the videos on CATER and less than 20% on CATER-h. This is because the tracker is unable to maintain object permanence through occlusions and containments, showcasing the challenging nature of the task. Our Hungarian tracking has 46.0% Top-1 on CATER and 37.2% Top-1 on CATER-h. On CATER, TPN and TSM, as two state-of-the-art methods focusing on temporal modeling for videos, achieve a higher accuracy than methods in Girdhar & Ramanan (2020). However, on CATER-h, TPN only has 50.2% Top-1 accuracy and 44.0% for TSM. SINet performs poorly even though SINet reasons about the higher-order object interactions via multi-headed attention and fusion of both coarse- and fine-grained information (similar to our **Hopper**). The poor performance of SINet can be attributed to the less accurate object representations and the lack of tracking whereas effective temporal modeling is critical for the task. Without the object-centric modeling, Transformer that uses a sequence of frame representations performs poorly. However, for both SINet and Transformer, a significant improvement is observed after utilizing our

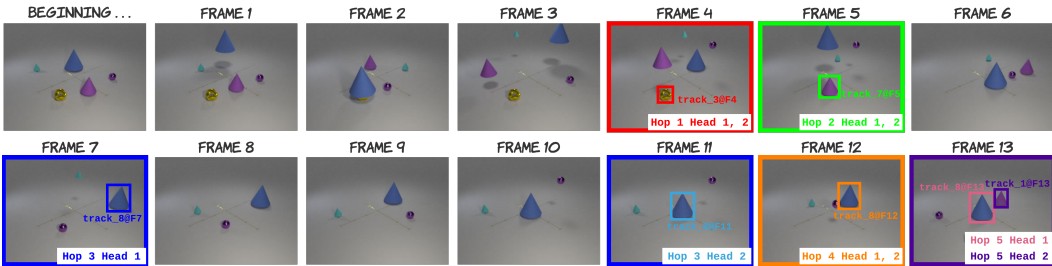

Snitch is contained by the purple cone at frame 5 (the last visible snitch is in frame 4). Then, purple cone is contained by the blue cone at frame 7. After that the blue cone keeps sliding. The blue cone uncontains the purple cone at frame 13. In the end, *the snitch is inside the purple cone*.

**Figure 3:** Qualitative result & interpretability of our model. We highlight the object attended per hop and per head from **Hopper**-multihop. The frame border of attended object is colored based on the hop index (accordingly: **Hop1**, **Hop2**, **Hop3**, **Hop4**, and **Hop5**). The bounding box of the *most* attended object in each hop shares the same color as the color of the hop index. Please zoom in to see the details. Best viewed in color.

**Hopper** framework. This shows the benefits of tracking-enabled object-centric learning embedded in **Hopper**. Since almost 60% of videos in CATER have the last visible snitch in the last frame, we run another experiment by using Transformer in **Hopper** but only uses the representations of the frame and objects *in the last frame*. This model variant achieves 61.1% Top-1 on CATER, even higher than the best-performing method in Girdhar & Ramanan (2020), but only 41.8% Top-1 on CATER-h, which proves and reiterates the necessity of CATER-h. The accuracy is relatively high, likely due to other dataset biases. More discussion is available in Appendix G.6. Our method, **Hopper** with the MHT, outperforms all baselines in Top-1 using only 13 frames per video, highlighting the effectiveness and importance of the proposed multi-hop reasoning. Shamsian et al. (2020) report slightly better Top-1 (74.8%) and $L_1$ (0.54) on CATER but they impose strong domain knowledge, operate at 24 FPS (300 frames per video) and require the location of the snitch to be labeled in every frame, even when contained or occluded, for *both* object detector and their reasoning module. Labeling these non-visible objects for every frame of a video would be very difficult in real applications. Furthermore, their method only has 51.4% Top-1 and $L_1$ of 1.31 on CATER-h at 24 FPS. Please refer to Appendix where we provide more quantitative results.

**Interpretability.** We visualize the attended objects in Figure 3. As illustrated, the last visible snitch is in frame 4 in this video, and at frame 5 snitch is contained by the purple cone. At frame 7, the purple cone is contained by the blue cone, but in the end, the blue cone excludes the purple cone. Hop 1 of **Hopper**-multihop attends to the last visible snitch at frame 4, and hop 2 attends to snitch's immediate container: the purple cone at frame 5. Hop 3 mostly attends to the purple cone's immediate container: the blue cone at frame 7, and secondly attends to the blue cone at frame 11 (by the other head) as the blue cone is just sliding from frame 7 till 12. Hop 4 mostly attends to the blue cone at frame 12. Hop 5 mostly attends to the purple cone (who contains the snitch) at frame 13, and secondly attends to the blue cone at frame 13. The visualization exhibits that **Hopper**-multihop performs reasoning by hopping over frames and meanwhile selectively attending to objects in the frame. It also showcases that MHT provides more transparency to the reasoning process. Moreover, MHT implicitly learns to perform snitch-oriented tracking automatically. More visualizations are available in Appendix.

## 7    CONCLUSION AND FUTURE WORK

This work presents **Hopper** with a novel Multi-hop Transformer to address object permanence in videos. **Hopper** achieves 73.2% Top-1 accuracy at just 1 FPS on CATER, and demonstrates the benefits of multi-hop reasoning. In addition, the proposed Multi-hop Transformer uses an iterative attention mechanism and produces a step-by-step reasoning chain that improves interpretability. Multi-hop models are often difficult to train without supervision for the middle hops. We propose several training methods that can be applied to other tasks to address the problem of lacking a ground truth reasoning chain. In the future, we plan to experiment on real-world video datasets and extend our methods to deal with other complex tasks (such as video QA).

ACKNOWLEDGMENTS

The research was supported in part by NSF awards: IIS-1703883, IIS-1955404, and IIS-1955365.

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

# A  ABLATION STUDY

| Dynamic Stride | Min 5 Hops | Hop 1 Loss | Hop 2 Loss | Frame Loss | Teacher Forcing | Debias Loss | Top 1 ↑ | Top 5 ↑ | $L_1 \downarrow$ |
|:---:|:---:|:---:|:---:|:---:|:---:|:---:|:---:|:---:|:---:|
| ✓ | ✓ | ✓ | ✓ | ✓ | ✓ | ✓ | 68.41 | 87.89 | 1.09 |
| ✓ | ✓ | ✓ | ✓ | ✓ | ✓ | ✗ | 64.65 | 87.75 | 1.19 |
| ✓ | ✓ | ✓ | ✓ | ✓ | ✗ | ✗ | 63.32 | 86.94 | 1.23 |
| ✓ | ✓ | ✓ | ✓ | ✗ | ✗ | ✗ | 62.88 | 86.57 | 1.19 |
| ✓ | ✓ | ✓ | ✗ | ✗ | ✗ | ✗ | 61.92 | 88.19 | 1.20 |
| ✓ | ✓ | ✗ | ✗ | ✗ | ✗ | ✗ | 59.48 | 84.58 | 1.37 |
| ✓ | ✗ | ✗ | ✗ | ✗ | ✗ | ✗ | 59.11 | 86.20 | 1.37 |
| ✗ | ✗ | ✗ | ✗ | ✗ | ✗ | ✗ | 57.27 | 84.43 | 1.39 |

**Table 3:** Ablation Study of `Hopper` training methods. We gradually add training methods described in Section 4, i.e., dynamic hop stride, minimal 5 hops of reasoning, auxiliary hop 1 object loss, auxiliary hop 2 object loss, auxiliary frame loss, teacher forcing, and contrastive debias loss via masking out, onto the base `Hopper`-multihop model. The results are obtained from the CATER-h test set.

We conduct an ablation study of training methods described in Section 4 in Table 3. As shown, all proposed training methods are beneficial. 'Dynamic Stride' gives the model more flexibility whereas 'Min 5 Hops' constrains the model to perform a reasonable number of steps of reasoning. 'Hop 1 Loss', 'Hop 2 Loss', 'Frame Loss' and 'Teacher Forcing' stress the importance of the correctness of the first 2 hops to avoid error propagation. 'Debias Loss' is the most effective one by contrastively inducing the latent space to capture information that is maximally useful to the task at hand.

| Object Detector & Tracker | Reasoning Model | Top 1 ↑ | Top 5 ↑ | $L_1 \downarrow$ |
|:---:|:---:|:---:|:---:|:---:|
| DETR + Hungarian (ours) | MHT (both Masked) | 65.73 | 88.39 | 1.13 |
| DETR + Hungarian (ours) | MHT (no Gating) | 66.62 | 87.43 | 1.16 |
| DETR (no Tracking) | MHT (no Tracking) | 67.51 | 88.32 | 1.14 |
| DETR + Hungarian (ours) | MHT (mask out LAST) | 32.49 | 60.92 | 2.53 |
| DETR + Hungarian (ours) | MHT (mask out ALL) | 11.68 | 28.98 | 3.60 |

**Table 4:** Ablation study & comparative results of analyzing components of our method (on CATER-h test set).

We then study how would different choices of the sub-components of our method affect the Snitch Localization performance. In Table 4:

- 'MHT (both Masked)': This refers to using our `Hopper`-multihop but replacing the original $U$ input to Transformer$_f$ with a masked version. In this way, both Transformer$_f$ and Transformer$_s$ have 'Masking()' applied beforehand.

- 'MHT (no Gating)': This refers to using our `Hopper`-multihop but removing the *Attentional Feature-based Gating* (line 11 in Algorithm 1) inside of MHT.

- 'MHT (no Tracking)': This refers to using our `Hopper`-multihop but entirely removing the Hungarian tracking module. Thus, MHT directly takes in unordered object representations as inputs.

- 'MHT (mask out LAST)': This refers to taking our trained `Hopper`-multihop, masking out the representation of the most attended object in the last hop by zeros, and then making predictions. This is to verify whether the most attended object in the last hop is important for the final Snitch Localization prediction task.

- 'MHT (mask out ALL)': Similar to the above, 'MHT (mask out ALL)' refers to taking our trained `Hopper`-multihop, masking out the representations of the most attended objects in *all* hops by zeros, and then making predictions. This is to verify how important are the most attended objects in all hops that are identified by our `Hopper`-multihop.

As shown in Table 4, all of these ablations give worse performance, thus, indicating that our motivations for these designs are reasonable (see Section 4). Recall that in Table 2, 'DETR + Hungarian' (without MHT) has only 37.2% Top-1 accuracy on CATER-h (learning a perfect object detector or tracker is not the focus of this paper). This highlights the superiority of our MHT as a reasoning model, and suggests that MHT has the potential to correct mistakes from the upstream object detector and tracker, by learning more robust object representations during the process of learning the Snitch Localization task. Masking out the most attended object identified by our **Hopper**-multihop in the last hop only has 32.49% Top-1 accuracy. Masking out all of the most attended objects from all hops only has 11.68% Top-1 accuracy. Such results reassure us about the interpretability of our method.

## B  Parameter Comparison

| Model | # Parameters (M) | GFLOPs | Top 1 Acc. |
|---|---|---|---|
| SINet | 138.69 | 7.98 | 18.6 |
| Transformer | 15.01 | 0.11 | 11.6 |
| **Hopper**-transformer (last frame) | 15.01 | 0.09 | 41.8 |
| **Hopper**-transformer | 15.01 | 1.10 | 57.6 |
| **Hopper**-sinet | 139.22 | 8.05 | 62.8 |
| **Hopper**-multihop (our proposed method) | 6.39 | 1.79 | 68.4 |

**Table 5:** Parameter and FLOPs comparison of our **Hopper**-multihop to alternative methods. (M) indicates millions. Results of the methods on CATER-h test set is also listed.

In Table 5, we compare the number of parameters of our **Hopper**-multihop with alternative methods. Our proposed method is the most efficient one in terms of the number of parameters. This is because of the iterative design embedded in MHT. Unlike most existing attempts on using Transformer that stack multiple encoder and decoder layers in a traditional way, MHT only has one layer of Transformer$_f$ and Transformer$_s$. As multiple iterations are applied to MHT, parameters of Transformer$_f$ and Transformer$_s$ *from different iterations* are shared. This iterative transformer design is inspired by previous work Locatello et al. (2020). This design saves parameters, accumulates previously learned knowledge, and adapts to varying number of hops, e.g., some require 1 hop and some require more than 1 hop (e.g, 5 hops). Because for these videos that tentatively only require 1 hop, stacking multiple layers of Transformer such as 5 might be wasteful and not necessary, our design of MHT could address such issue, being more parameter-efficient. We also report the GFLOPs comparison. Given that the FLOPs for MHT depends on the number of hops predicted for a video, we report the average number of FLOPs for the CATER-h test set.

## C  Diagnostic Analysis
### C.1  Diagnostic Analysis on the Hopping Mechanism

| # Hops | 1 | 2 | 3 | 4 | $\geq 5$ |
|---|---|---|---|---|---|
| Ground Truth | 104 | 105 | 111 | 110 | 1026 |
| Prediction | 109 | 102 | 112 | 108 | 1025 |
| Jaccard Similarity $\uparrow$ | 0.9541 | 0.9714 | 0.9561 | 0.9818 | 0.9990 |

**Table 6:** Diagnostic analysis of the Multi-hop Transformer in terms of the 'hopping' ability (# hops performed).

We evaluate the 'hopping' ability of the proposed Multi-hop Transformer in Table 6. The prediction is made by our **Hopper**-multihop that requires at least 5 hops of reasoning unless not possible. For each test video, we compute the ground truth number of hops required by this video and obtain the number of hops that **Hopper** actually runs. In the table, we provide the ground truth count and predicted count of test videos that require 1 hop, 2 hops, 3 hops, 4 hops, and equal or greater than 5 hops. Since the numbers are close, we further compute Jaccard Similarity (range from 0 to 1 and higher is better) to measure the overlapping between the ground truth set of the test videos and predicted set of the test videos. According to these metrics, our proposed **Hopper**-multihop functionally performs the correct number of hops for almost all test videos.

| Frame Index | 1 | 2 | 3 | 4 | 5 | 6 | 7 | 8 | 9 | 10 | 11 | 12 | 13 |
|---|---|---|---|---|---|---|---|---|---|---|---|---|---|
| Hop 1 | 127 | 136 | 106 | 120 | 104 | 111 | 109 | 107 | 105 | 0 | 0 | 0 | 0 |
| Hop 2 | 0 | 127 | 136 | 106 | 120 | 104 | 111 | 109 | 107 | 105 | 0 | 0 | 0 |
| Hop 3 | 0 | 0 | 8 | 2 | 66 | 10 | 13 | 24 | 17 | 107 | 778 | 0 | 0 |
| Hop 4 | 0 | 0 | 0 | 1 | 0 | 0 | 0 | 1 | 0 | 0 | 10 | 1013 | 0 |
| Hop 5 | 0 | 0 | 0 | 0 | 0 | 0 | 0 | 0 | 0 | 0 | 0 | 9 | 1016 |
| Hop 6 | 0 | 0 | 1 | 0 | 0 | 0 | 1 | 0 | 0 | 0 | 0 | 0 | 9 |

**Table 7:** Hop Index vs. Frame Index: the number of times hop $h$ mostly attends to frame $t$ (results are obtained from **Hopper**-multihop on CATER-h). See details in Appendix C.1.

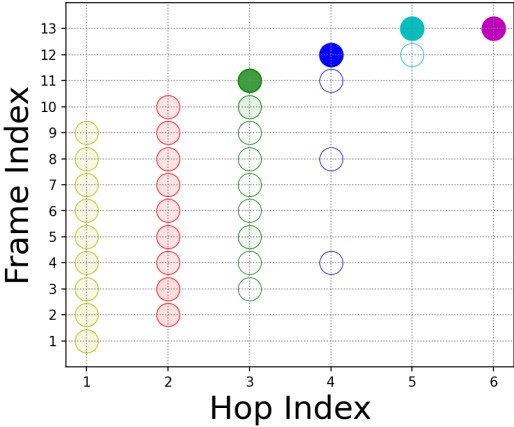

**Figure 4:** Hop Index vs. Frame Index: we plot the index of frame of the most attended object identified by each hop. Each hop has its unique color, and the transparency of the dot denotes the normalized frequency of that frame index for that particular hop.

In Table 7, we show the number of times hop $h$ mostly attends to frame $t$. The results are obtained from **Hopper**-multihop on CATER-h, for those $1025$ test videos predicted with $\geq 5$ hops shown in Table 6. In Figure 4, we plot the index of frame of the most attended object identified by each hop (conveying the same meaning as Table 7). The transparency of the dot denotes the normalized frequency of that frame index for that particular hop.

We can observe that: **(1)** Hop $3$ to $6$ tend to attend to later frames, and this is due to lacking supervision for the intermediate hops. As we discussed in Section 5, multi-hop model is hard to train in general when the ground truth reasoning chain is missing during training (Dua et al., 2020; Chen & Durrett, 2018; Jiang & Bansal, 2019; Wang et al., 2019a). Researchers tend to use ground truth reasoning chain as supervision when they train a multi-hop model (Qi et al., 2019; Ding et al., 2019; Chen et al., 2019). The results reconfirm that, without supervision for the intermediate steps, it is not easy for a model to automatically figure out the ground truth reasoning chain; **(2)** MHT has learned to predict the next frame of the frame that is identified by hop $1$, as the frame that hop $2$ should attend to; **(3)** there are only $9$ videos predicted with more than $5$ hops even though we only constrain the model to perform *at least* $5$ hops (unless not possible). Again, this is because no supervision is provided for the intermediate hops. As the Snitch Localization task itself is largely focused on the last frame of the video, without supervision for the intermediate hops, the model tends to "look at" later frames as soon as possible. These results suggest where we can improve for the current MHT, e.g., one possibility is to design self-supervision for each intermediate hop.

## C.2 COMPARATIVE DIAGNOSTIC ANALYSIS ACROSS FRAME INDEX

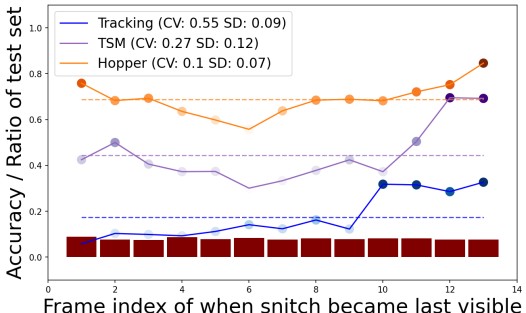

**Figure 5:** Diagnostic analysis of the performance in terms of when snitch becomes last visible in the video.

In Figure 5, we present the comparative diagnostic analysis of the performance in terms of when snitch becomes last visible. We bin the test set using the frame index of when snitch becomes last visible in the video. For each, we show the test set distribution with the bar plot, the performance over that bin using the line plot, and performance of that model on the full test set with the dashed line. We find that for Tracking (DaSiamRPN) and TSM, the Snitch Localization performance drops if the snitch becomes not visible earlier in the video. Such phenomenon, though still exists, but is alleviated for **Hopper**-multihop. We compute the standard deviation (SD) and coefficient of

variation (CV). Both are measures of relative variability. The higher the value, the greater the level of dispersion around the mean. The values of these metrics as shown in Figure 5 further reinforce the stability of our model and necessity of CATER-h dataset.

## D    EXTRA QUALITATIVE RESULTS

In Figure 6, we visualize the attention weights per hop and per head from Transformer$_s$ to showcase the hops performed by **Hopper**-multihop for video 'CATERh_054110' (the one in Figure 3) in details. Please see Figure 7, 8, 9, 10, and 11 for extra qualitative results from **Hopper**-multihop. We demonstrate the reasoning process for different cases (i.e., 'visible', 'occluded', 'contained', 'contained recursively', and 'not visible very early in the video').

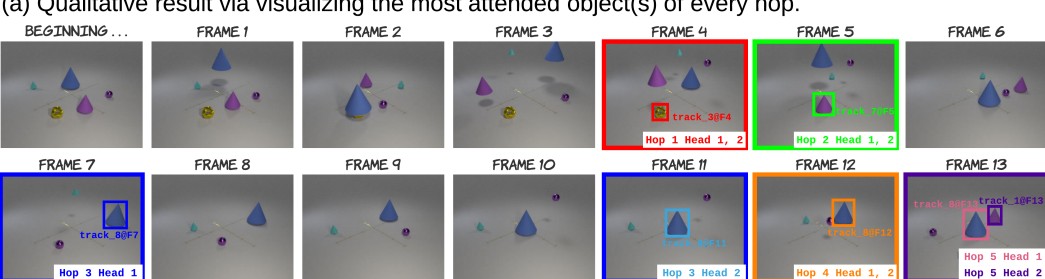

(a) Qualitative result via visualizing the most attended object(s) of every hop.

```
Snitch is contained by the purple cone at frame 5 (the last visible snitch is in frame 4). Then, purple
cone is contained by the blue cone at frame 7. After that the blue cone keeps sliding. The blue cone
uncontains the purple cone at frame 13. In the end, the snitch is inside the purple cone.
```

(b) Attention weights visualization.

```
Hop 1: Video CATERh_054110 mostly attends to the 30th element from the object tracks in the memory sequence, i.e., track 3 at frame 4
Hop 1 Head 1: Video CATERh_054110 mostly attends to the 30th element from the object tracks in the memory sequence, i.e., track 3 at frame 4
Hop 1 Head 2: Video CATERh_054110 mostly attends to the 30th element from the object tracks in the memory sequence, i.e., track 3 at frame 4

Hop 2: Video CATERh_054110 mostly attends to the 83th element from the object tracks in the memory sequence, i.e., track 7 at frame 5
Hop 2 Head 1: Video CATERh_054110 mostly attends to the 83th element from the object tracks in the memory sequence, i.e., track 7 at frame 5
Hop 2 Head 2: Video CATERh_054110 mostly attends to the 83th element from the object tracks in the memory sequence, i.e., track 7 at frame 5

Hop 3: Video CATERh_054110 mostly attends to the 98th element from the object tracks in the memory sequence, i.e., track 8 at frame 7
Hop 3 Head 1: Video CATERh_054110 mostly attends to the 98th element from the object tracks in the memory sequence, i.e., track 8 at frame 7
Hop 3 Head 2: Video CATERh_054110 mostly attends to the 102th element from the object tracks in the memory sequence, i.e., track 8 at frame 11

Hop 4: Video CATERh_054110 mostly attends to the 103th element from the object tracks in the memory sequence, i.e., track 8 at frame 12
Hop 4 Head 1: Video CATERh_054110 mostly attends to the 103th element from the object tracks in the memory sequence, i.e., track 8 at frame 12
Hop 4 Head 2: Video CATERh_054110 mostly attends to the 103th element from the object tracks in the memory sequence, i.e., track 8 at frame 12

Hop 5: Video CATERh_054110 mostly attends to the 13th element from the object tracks in the memory sequence, i.e., track 1 at frame 13
Hop 5 Head 1: Video CATERh_054110 mostly attends to the 104th element from the object tracks in the memory sequence, i.e., track 8 at frame 13
Hop 5 Head 2: Video CATERh_054110 mostly attends to the 13th element from the object tracks in the memory sequence, i.e., track 1 at frame 13
```

**Figure 6:** Visualization of attention weights & interpretability of our model. In (a), we highlight object(s) attended in every hop from **Hopper**-multihop (frame border is colored accordingly: **Hop1**, **Hop2**, **Hop3**, **Hop4**, and **Hop5**). In (b), we visualize the attention weights per hop (the smaller attention weight that an object has, the larger opacity is plotted for that object entity). As shown, **Hopper**-multihop performs 5 hops of reasoning for the video 'CATERh_054110'. Our model performs reasoning by hopping over frames and meanwhile selectively attending to objects in the frame. Please zoom in to see the details. Best viewed in color.

## E    TRACK REPRESENTATION VISUALIZATION

Please see Figure 12 for visualization of the object track representations of video 'CATERh_054110' (attention weights from **Hopper**-multihop for this video are shown in Figure 6). **Hopper** utilizes tracking-integrated object representations since tracking can link object representations through time and the resulting representations are more informative and consistent. As shown in the figure, the tracks that are obtained from our custom Hungarian algorithm that are competitive. Our model **Hopper**-multihop takes in the best-effort object track representations (along with the coarse-grained frame track) as the source input to the Multi-hop Transformer, and then further learns the most useful and correct task-oriented track information implicitly (as shown in Figure 6).

(a) Qualitative result via visualizing the most attended object(s) of every hop.

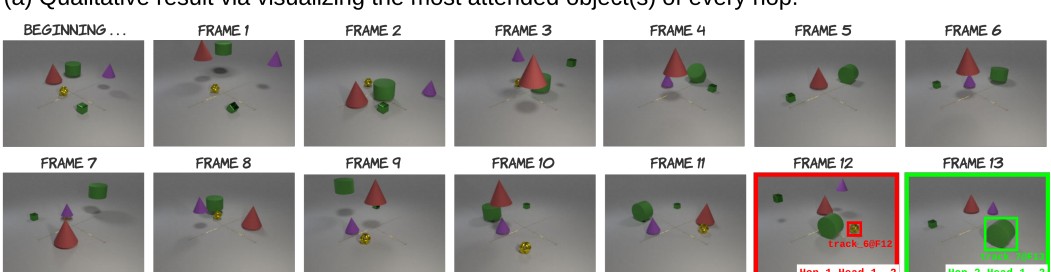

(Visible) In the end, ***the snitch is visible***.

(b) Attention weights visualization.

**Figure 7:** We visualize the attention weights per hop and per head from Transformer$_s$ in our **Hopper**-multihop. Objects attended in every hop are highlighted (whose frame border is colored accordingly: **Hop1**, **Hop2**, **Hop3**, **Hop4**, and **Hop5**). Please zoom in to see the details. Best viewed in color.

(a) Qualitative result via visualizing the most attended object(s) of every hop.

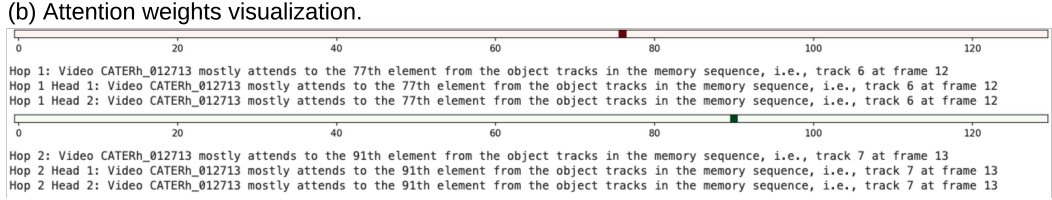

(Occlusion) In the end, ***the snitch is occluded by the green cylinder***.

(b) Attention weights visualization.

**Figure 8:** We visualize the attention weights per hop and per head from Transformer$_s$ in our **Hopper**-multihop. Objects attended in every hop are highlighted (whose frame border is colored accordingly: **Hop1**, **Hop2**, **Hop3**, **Hop4**, and **Hop5**). Please zoom in to see the details. Best viewed in color.

## F  FAILURE CASES

We present a sample of the failure cases of **Hopper**-multihop in Figure 15. Generally, a video is more difficult if: (1) there are similar looking objects present simultaneously (especially if the object is similar to snitch or another cone in the video); (2) wrong hop 1 or 2 identified; (3) critical moments are occluded (e.g. in Figure 15, when the snitch becomes occluded, it is contained by the brown cone); (4) complex object interactions such as recursive containment along with container moving (since such case usually has the last visible snitch very early). **Hopper**-multihop fails in the first scenario due to the error made by the object representation and detection module, which can be avoided by using a fine-tuned object detector model. **Hopper**-multihop fails in the second scenario can attribute to the error made by the object detector, tracker, our heuristics, or capability of the inadequately-trained Multi-hop Transformer. The third scenario is not easy even for humans under 1 FPS, thus increasing FPS with extra care might ease the problem. The last scenario requires more sophisticated multi-step reasoning, thus changing the minimal number of hops of the Multi-hop

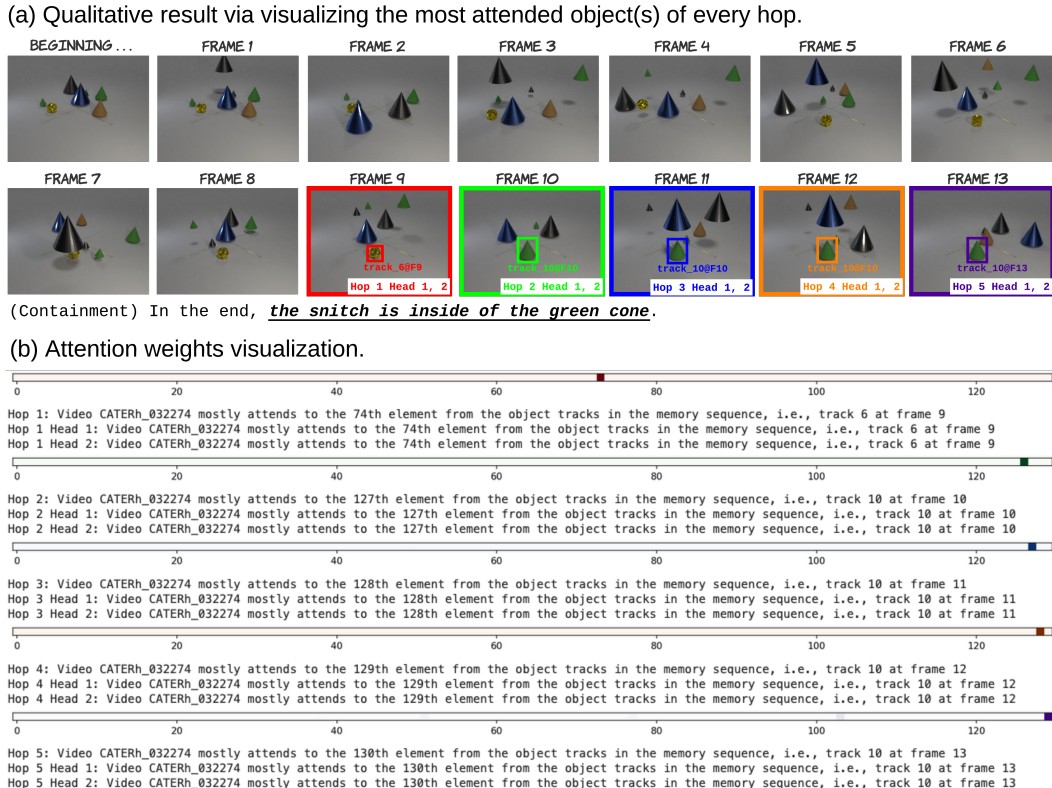

(a) Qualitative result via visualizing the most attended object(s) of every hop.

(Containment) In the end, ***the snitch is inside of the green cone***.

(b) Attention weights visualization.

**Figure 9:** We visualize the attention weights per hop and per head from Transformer$_s$ in our **Hopper**-multihop. Objects attended in every hop are highlighted (whose frame border is colored accordingly: **Hop1**, **Hop2**, **Hop3**, **Hop4**, and **Hop5**). Please zoom in to see the details. Best viewed in color.

Transformer into a larger number with self-supervision for the intermediate hops to handle the long hops should help in solving this scenario. Overall, an accurate backbone, object detector, tracking method, or the heuristics to determine visibility and the last visible snitch's immediate container (or occluder) will help improve the performance of **Hopper**-multihop. We would like to focus on enhancing **Hopper**-multihop for these challenges and verify our hypothesis in our future work.

## G  THE CATER-H DATASET

### G.1  BASICS: CATER

CATER (Girdhar & Ramanan, 2020) provides a diagnostic video dataset that requires spatial and temporal understanding to be solved. It is built against models that take advantage of wrong scene biases. With fully observable and controllable scene bias, the $5,500$ videos in CATER are rendered synthetically at 24 FPS (300-frame 320x240px) using a library of standard 3D objects: 193 different object classes in total which includes 5 object shapes (cube, sphere, cylinder, cone, snitch) in 3 sizes (small, medium, large), 2 materials (shiny metal and matte rubber) and 8 colors. Every video has a small metal snitch (see Figure 1). There is a large "table" plane on which all objects are placed. At a high level, the dynamics in CATER videos are in analogy to the cup-and-balls magic routine[4]. A subset of 4 atomic actions ('rotate', 'pick-place', 'slide' and 'contain') is afforded by each object. See Appendix G.2 for definition of the actions. Note that 'contain' is only afforded by cone and recursive containment is possible, i.e., a cone can contain a smaller cone that contains another object. Every video in CATER is split into several time slots, and every object in this video randomly performs an action in the time slot (including 'no action'). Objects and actions vary across videos. The "table" plane is divided into $6 \times 6$ grids (36 rectangular cells), and the *Snitch*

---

[4]https://en.wikipedia.org/wiki/Cups_and_balls

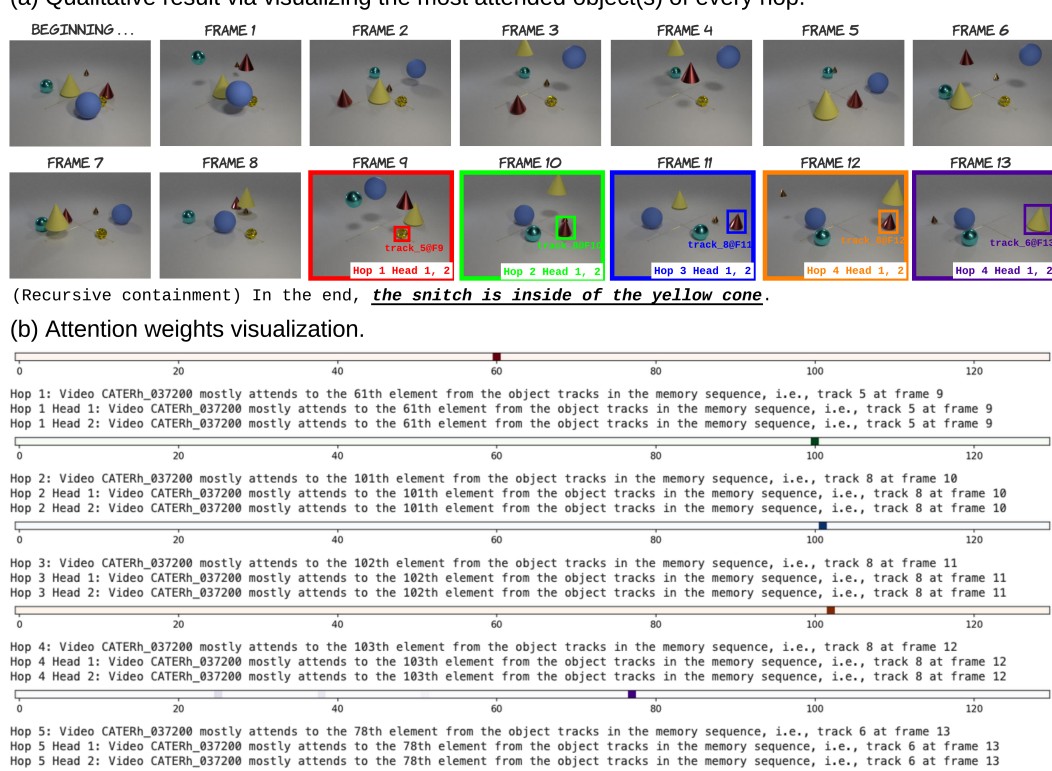

(a) Qualitative result via visualizing the most attended object(s) of every hop.

(b) Attention weights visualization.

**Figure 10:** We visualize the attention weights per hop and per head from Transformer$_s$ in our **Hopper**-multihop. Objects attended in every hop are highlighted (whose frame border is colored accordingly: **Hop1**, **Hop2**, **Hop3**, **Hop4**, and **Hop5**). Please zoom in to see the details. Best viewed in color.

*Localization* task is to determine the grid that the snitch is in at the end of the video, as a single-label classification task. The task implicitly requires the understanding of object permanence because objects could be occluded or contained (hidden inside of) by another object.

## G.2  DEFINITION OF ACTIONS

We follow the definition of the four atomic actions in Girdhar & Ramanan (2020). Specifically:

1. **'rotate':** the 'rotate' action means that the object rotates by $90°$ about its perpendicular or horizontal axis, and is afforded by cubes, cylinders and the snitch.

2. **'pick-place':** The 'pick-place' action means the object is picked up into the air along the perpendicular axis, moved to a new position, and placed down. This is afforded by all objects.

3. **'slide':** the 'slide' action means the object is moved to a new location by sliding along the bottom surface, and is also afforded by all objects.

4. **'contain':** 'contain' is a special operation, only afforded by the cones, in which a cone is pick-placed on top of another object, which may be a sphere, a snitch or even a smaller cone. This allows for recursive containment, as a cone can contain a smaller cone that contains another object. Once a cone 'contains' an object, the 'slide' action of the cone effectively slides all objects contained within the cone. This holds until the top-most cone is pick-placed to another location, effectively ending the containment for that top-most cone.

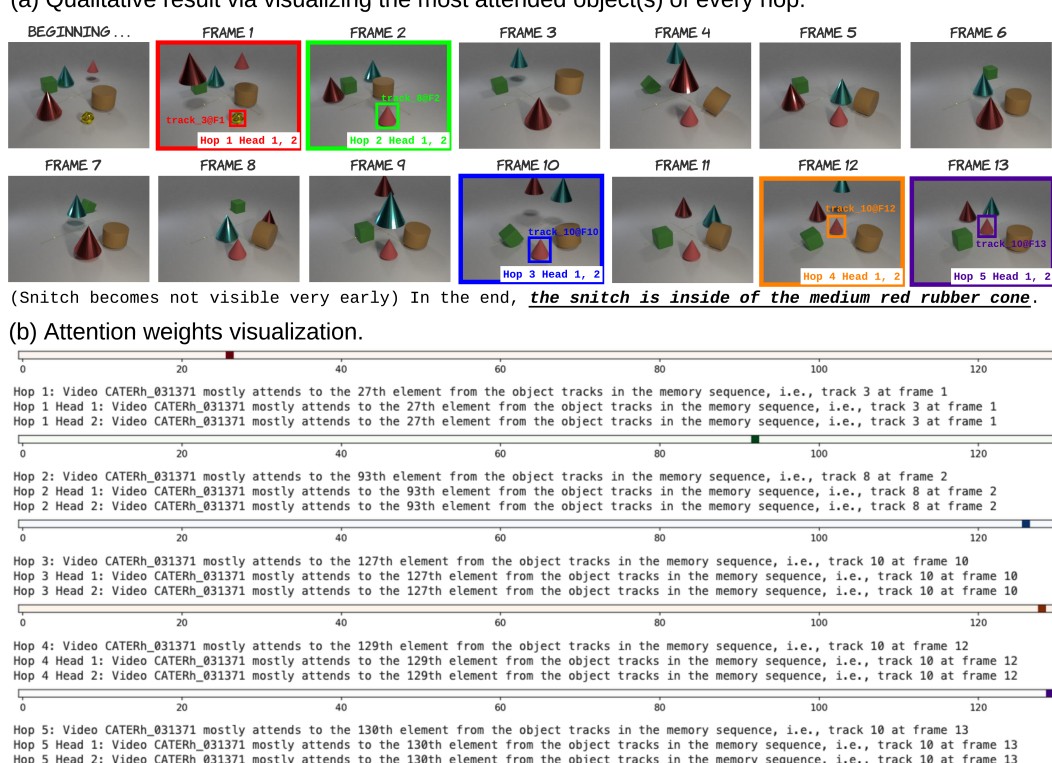

(a) Qualitative result via visualizing the most attended object(s) of every hop.

(Snitch becomes not visible very early) In the end, *the snitch is inside of the medium red rubber cone*.

(b) Attention weights visualization.

**Figure 11:** We visualize the attention weights per hop and per head from Transformer$_s$ in our **Hopper**-multihop. Objects attended in every hop are highlighted (whose frame border is colored accordingly: **Hop1**, **Hop2**, **Hop3**, **Hop4**, and **Hop5**). Please zoom in to see the details. Best viewed in color.

### G.3 DATASET GENERATION PROCESS

The generation of the CATER-h dataset is built upon the CLEVR (Johnson et al., 2017a) and CATER (Girdhar & Ramanan, 2020) codebases. Blender is used for rendering. The animation setup is the same as the one in CATER. A random number of objects with random parameters are spawned at random locations at the beginning of the video. They exist on a $6 \times 6$ portion of a 2D plane with the global origin in the center. Every video has a snitch, and every video is split into several time slots. Each action is contained within its time slot. At the beginning of each slot, objects are randomly selected to perform a random action afforded by that object (with no collision ensured). Please refer to Girdhar & Ramanan (2020) for more animation details.

In order to have a video dataset that emphasizes on recognizing the effect of the temporal variations on the state of the world, we set roughly equal number of video samples to have the last visible snitch along the temporal axis. In order to obtain such a dataset, we generated a huge number of videos, computed the frame index of the last visible snitch in every video under 1 FPS (13 frames per video). Then, for every frame index $i$, we obtained the set of videos whose last visible snitch is at frame index $i$, and finally, we randomly chose $500$ and more videos from this set and discarded the rest. Eventually, the total number of videos in CATER-h is $7,080$. We split the data randomly in $70 : 30$ ratio into a training and test set, resulting in $5,624$ training samples and $1,456$ testing samples.

### G.4 CATER-H V.S. CATER

Figure 13 compares the CATER-h dataset and the CATER dataset.

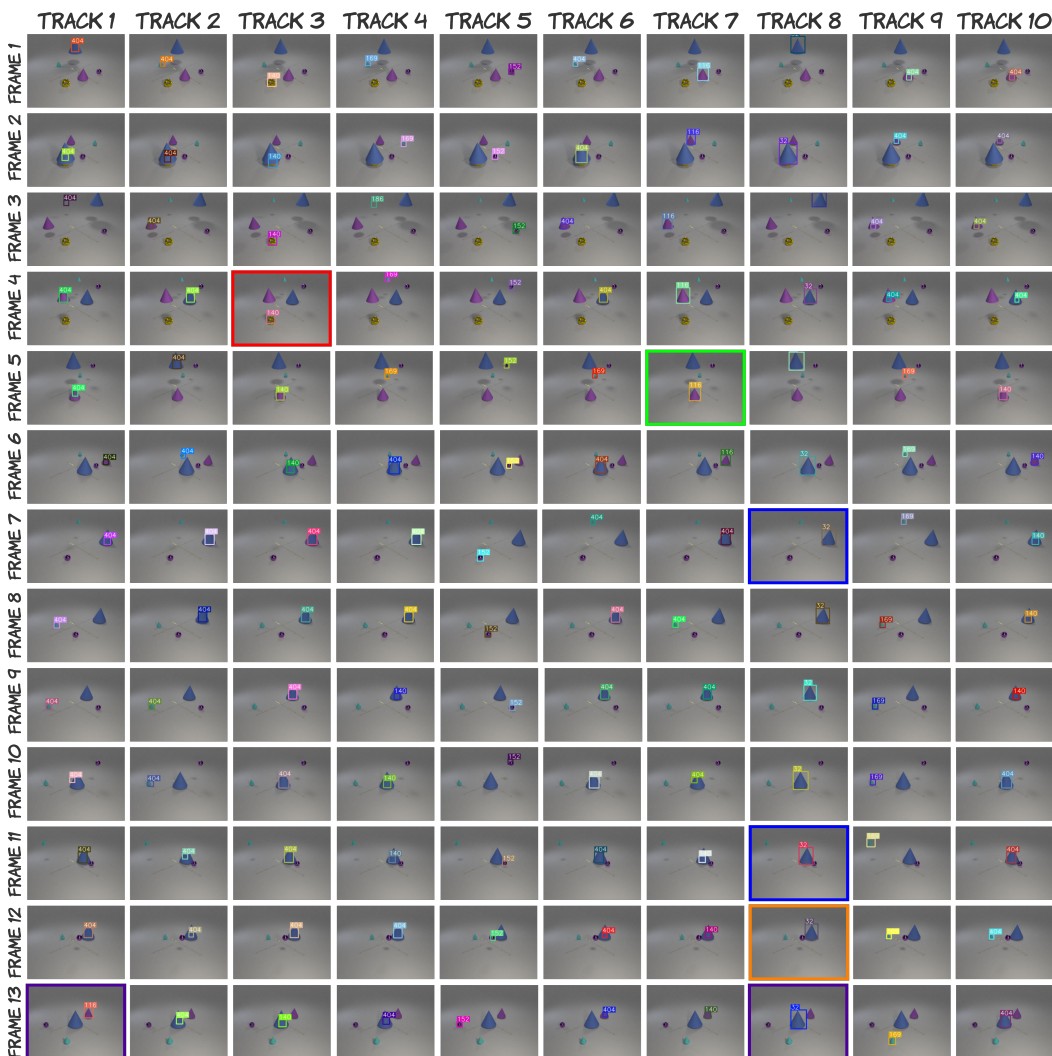

**Figure 12:** Tracking-integrated object representation visualization. `Hopper` utilizes tracking-integrated object representations since tracking can link object representations through time and the resulting representations are more informative and consistent. We visualize the object track representations of video 'CATERh_054110' (attention weights from `Hopper`-multihop for this video are shown in Figure 6). Here, every column is for a track and every row is for a frame. The bounding box and object class label computed from the object representation are plotted (404 is the object class label for ∅, i.e., none object, and 140 is the object class label for the snitch). As shown in the figure, the tracks that are obtained from our designed Hungarian algorithm are not perfect but acceptable since having perfect tracking here is not the goal of this paper. Our model `Hopper`-multihop takes in the (imperfect) object track representations (along with the coarse-grained frame track) as the source input to the Multi-hop Transformer, and then further learns the most useful and correct task-oriented track information implicitly (as shown in Figure 6). `Hopper`-multihop preforms 5 hops of reasoning for this video; objects attended in every hop are highlighted (whose frame border is colored accordingly: **Hop1**, **Hop2**, **Hop3**, **Hop4**, and **Hop5**). Please zoom in to see the details. Best viewed in color.

## G.5 TRAIN/TEST DISTRIBUTION

Figure 14 shows the data distribution over classes in CATER-h.

## G.6 OTHER POTENTIAL DATASET BIAS

As shown in Table 2, '`Hopper`-transformer (last frame)' still has a relatively high accuracy on CATER-h. We hypothesize that the reason it has $41.8\%$ Top-1 accuracy on CATER-h might be due

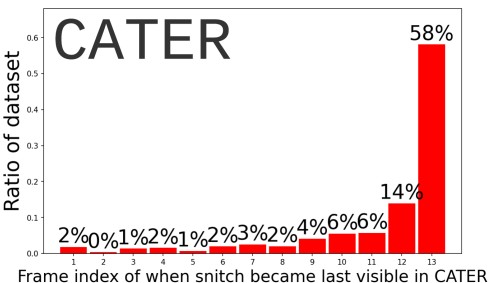 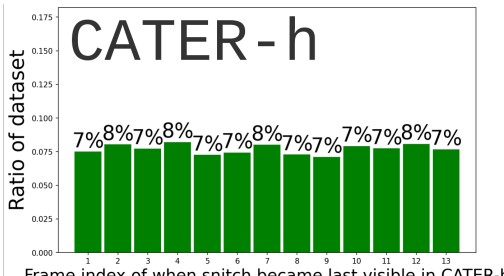

**Figure 13:** Histogram of the frame index of the last visible snitch of every video. We find that CATER is highly imbalanced for the Snitch Localization task in terms of the temporal cues: e.g., snitch is entirely visible at the end of the video for $58\%$ samples. This temporal bias results in high-accuracy even if it ignores all but the last frame of the video. Our dataset CATER-h addresses this issue with a balanced dataset.

to other dataset biases (apart from the snitch grid distribution bias, and temporal bias that CATER has). Upon further investigation, we identify one type of additional bias, the "cone bias", i.e., snitch can only be contained by a cone in the videos of CATER and CATER-h.

In order to verify the existence of the "cone bias", we compute the accuracy if we make a random guess among the grids of cones that are not covered by any other cones, for all test videos whose snitch is covered in the end. This gives us $48.26\%$ Top-1 accuracy. This shows that the "cone bias" does exist in the dataset. The so-called "cone bias" comes from the nature of objects used in CATER and the fact that only the cone can carry the snitch (thus, it is closer to a feature of the dataset, rather than being a "bias" per se). Furthermore, because of the animation rules of CATER, there might exist other dataset biases, such as bias in terms of object size and shape, etc., which are hard to discover and address. This highlights the glaring challenge in building a fully unbiased (synthetic or real) dataset. CATER-h addresses the temporal bias that CATER has. A model has to perform long-term spatiotemporal reasoning in order to have a high accuracy on CATER-h.

## H    IMPLEMENTATION DETAILS

### H.1    **HOPPER**

We introduce the implementation of **Hopper**-multihop in this section. For both training and testing, we only used 1 FPS (frames per second) to demonstrate the efficiency of our approach. This means we only have 13 frames per video. Note that we trained the video query representation and recognition part (Multi-hop Transformer along with the final MLP) end to end.

The CNN backbone we utilized is the pre-trained ResNeXt-101 (Xie et al., 2017) model from Ma et al. (2018). We trained DETR (Carion et al., 2020)[5] on LA-CATER (Shamsian et al., 2020) which is a dataset with generated videos following the same configuration to the one used by CATER, but additional ground-truth object bounding box location and class label annotations are available (Shamsian et al. (2020) predicts the bounding box of snitch in the video given the supervision of the bounding box of the snitch in 300 frames). We followed the settings in Carion et al. (2020) to set up and train DETR, e.g., stacking 6 transformer encoder layers and 6 transformer decoder layers, utilizing the object detection set prediction loss and the auxiliary decoding loss per decoder layer. $d$ is 256, $N$ is 10 and $C$ is 193. The initial $N$ object query embeddings are learned. The MLP for recognizing the object class label is one linear layer and for obtaining the object bounding box is a MLP with 2 hidden layers with $d$ neurons. After DETR was trained, we tested it on CATER to obtain the object representations, predicted bounding boxes and class labels. For tracking, we set $\lambda_c = 1, \lambda_b = 0$ because under a low FPS, using the bounding boxes for tracking is counterproductive, and it yielded reasonable results. Then, we trained the video recognition part, i.e., Multi-hop Transformer along with the final MLP, end to end with Adam (Kingma & Ba, 2014) optimizer. The final MLP we used is one linear layer that transforms the video query representation $\mathbf{e}$ of dimension $d$ into the grid class logits. The initial learning rate was set to $10^{-4}$ and weight decay to $10^{-3}$. The batch size was 16. The number of attention heads for DETR was set to 8 and

---

[5]https://github.com/facebookresearch/detr

for the Multi-hop Transformer was set to 2. Transformer dropout rate was set to $0.1$. We used multi-stage training with the training methods proposed. Moreover, we found that DETR tends to predict a snitch for every cone on the "table" plane when there is no visible snitch in that frame. To mitigate this particular issue of the DETR object detector trained on Shamsian et al. (2020), we further compute an object visibility map $\mathcal{V} \in \mathbb{R}^{NT \times 1}$, which is a binary vector and determined by a heuristic: an object is visible if the bounding box of the object is not completely contained by any bounding box of another object in that frame. The 'Masking()' function uses $\mathcal{V}$ by considering only the visible objects.

## H.2 Description of Reported CATER Baselines

For CATER, we additionally compare our results with the ones reported by Girdhar & Ramanan (2020) from TSN (Wang et al., 2016), I3D (Carreira & Zisserman, 2017), NL (Wang et al., 2018) as well as their LSTM variants. Specifically, TSN (Temporal Segment Networks) was the top performing method that is based on the idea of long-range temporal structure modeling before TSM and TPN. Two modalities were experimented with TSN, i.e., RGB or Optical Flow (that captures local temporal cues). I3D inflates 2D ConvNet into 3D for efficient spatiotemproal feature learning. NL (Non-Local Networks) proposed a spacetime non-local operation as a generic building block for capturing long-range dependencies for video classification. In order to better capture the temporal information for these methods, Girdhar & Ramanan (2020) further experimented with a 2-layer LSTM aggregation that operates on the last layer features before the logits. Conclusions from Girdhar & Ramanan (2020) are: (1) TSN ends up performing significantly worse than I3D instead of having similar performance which contrasts with standard video datasets; (2) the optical flow modality does not work well as the Snitch Localization task requires recognizing objects which is much harder from the optical flow; (3) more sampling from the video would give higher performance; (4) LSTM for more sophisticated temporal aggregation leads to a major improvement in performance.

## H.3 Baselines

We introduce the implementation of the baselines that we experimented with in this section. First, for our Hungarian tracking baseline, for every test video, we obtain the snitch track (based on which track's first object has snitch as its label) produced from our Hungarian algorithm, and project the center point of the bounding box of the last object in that track to the 3D plane (and eventually, the grid class label) by using a homography transformation between the image and the 3D plane (same method used in Girdhar & Ramanan (2020)). We also try with the majority vote, i.e., obtain the snitch track as the track who has the highest number of frames classified as snitch. We report the majority vote result in Table 1 and 2 because it is a more robust method. The results of using the first frame of our Hungarian tracking baseline are $31.8\%$ Top-1, $40.2\%$ Top-5, $2.7$ $L_1$ on CATER, and $28.2\%$ Top-1, $36.3\%$ Top-5, $2.8$ $L_1$ on CATER-h.

We used the public available implementation provided by the authors (Yang et al., 2020) for TSM and TPN. Models were defaultly initialized by pre-trained models on ImageNet (Deng et al., 2009). The original ResNet (He et al., 2016) serves as the 2D backbone, and the inflated ResNet (Feichtenhofer et al., 2019) as the 3D backbone network. We used the default settings of TSM and TPN provided by Yang et al. (2020), i.e., TSM-50 (2D ResNet-50 backbone) settings that they used to obtain results on Something-Something (Goyal et al., 2017), which are also the the protocols used in Lin et al. (2019), as well as TPN-101 (I3D-101 backbone, i.e. 3D ResNet-101) with multi-depth pyramid and the parallel flow that they used to obtain results on Kinetics (their best-performing setting of TPN) (Carreira & Zisserman, 2017). Specifically, the augmentation of random crop, horizontal flip and a dropout of $0.5$ were adopted to reduce overfitting. BatchNorm (BN) was not frozen. A momentum of $0.9$, a weight decay of $0.0001$ and a synchronized SGD with the initial learning rate $0.01$, which would be reduced by a factor of 10 at 75, 125 epochs (150 epochs in total). The weight decay for TSM was set to $0.0005$. TPN used auxiliary head, spatial convolutions in semantic modulation, temporal rate modulation and information flow (Yang et al., 2020). For SINet, we used the implementation provided by Ma et al. (2018). Specifically, image features for SINet were obtained from a pre-trained ResNeXt-101 (Xie et al., 2017) with standard data augmentation (randomly cropping and horizontally flipping video frames during training). Note that the image features used by SINet are the same as the ones used in our **Hopper**. The object

features were generated from a Deformable RFCN (Dai et al., 2017). The maximum number of objects per frame was set to 10. The number of subgroups of higher-order object relationships ($K$) was set to 3. SGD with Nesterov momentum were used as the optimizer. The initial learning rate was 0.0001 and would drop by 10x when validation loss saturates for 5 epochs. The weight decay was 0.0001 and the momentum was 0.9. The batch size was 16 for these 3 baselines. Transformer, **Hopper**-transformer, and **Hopper**-sinet used the Adam optimizer with a total of 150 epochs, a initial learning rate of $10^{-4}$, a weight decay of $10^{-3}$, and a batch size of 16. Same as our model, the learning rate would drop by a factor of 10 when there has been no improvement for 10 epochs on the validation set. The number of attention heads for the Transformer (and **Hopper**-transformer) was set to 2, the number of transformer layers was set to 5 to match the 5 hops in our Multi-hop Transformer, and the Transformer dropout rate was set to 0.1. For OPNet related experiments, we used the implementation provided from authors (Shamsian et al., 2020). We verified we could reproduce their results under 24 FPS on CATER by using their provided code and trained models.

For the Random baseline, it is computed as the average performance of random scores passed into the evaluation functions Girdhar & Ramanan (2020). For the Tracking baseline, we use the DaSiamRPN implementation from Girdhar & Ramanan (2020) [6]. Specifically, the ground truth information of the starting position of the snitch was first projected to screen coordinates using the render camera parameters. A fixed size box around the snitch is defined to initialize the tracker, and run the tracker until the end of the video. At the last frame, the center point of the tracked box is projected to the 2D plane by using a homography transformation between the image and the 2D plane, and then converted to the class label. With respect to TSN, I3D, NL and their variants, the results were from Girdhar & Ramanan (2020), and we used the same train, val split as theirs when obtaining our results on CATER.

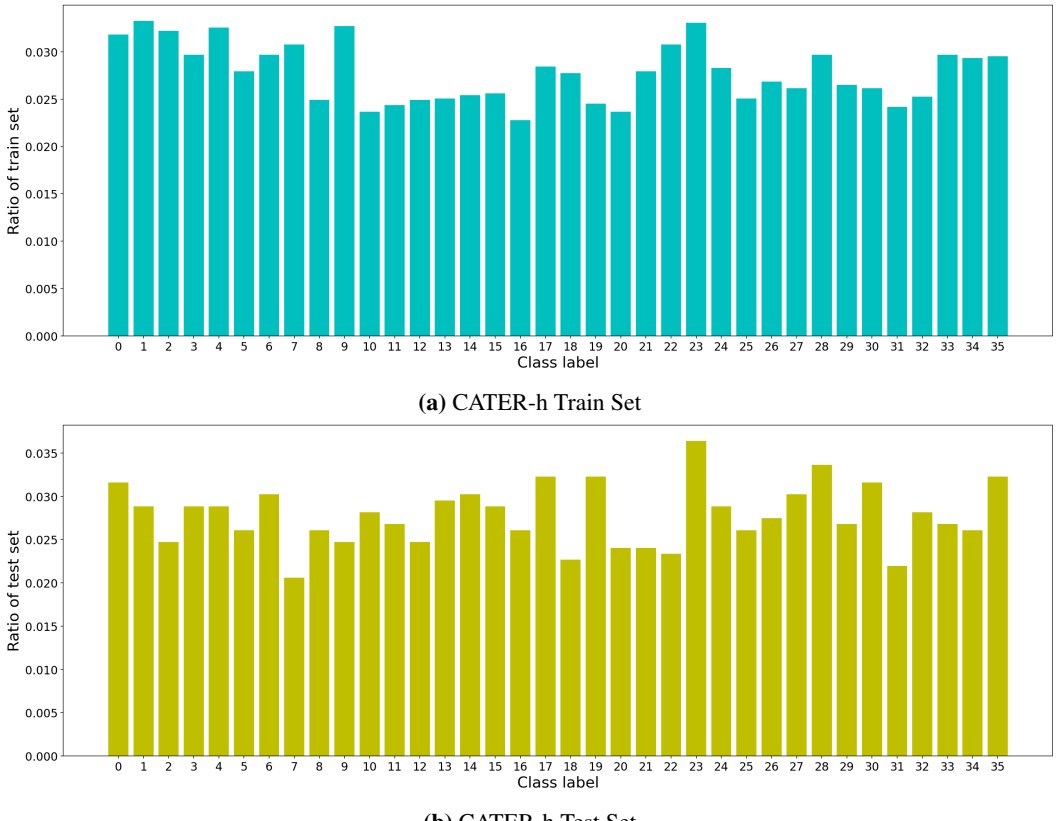

**(a)** CATER-h Train Set

**(b)** CATER-h Test Set

**Figure 14:** Data distribution over classes in CATER-h.

---

[6]https://github.com/rohitgirdhar/CATER

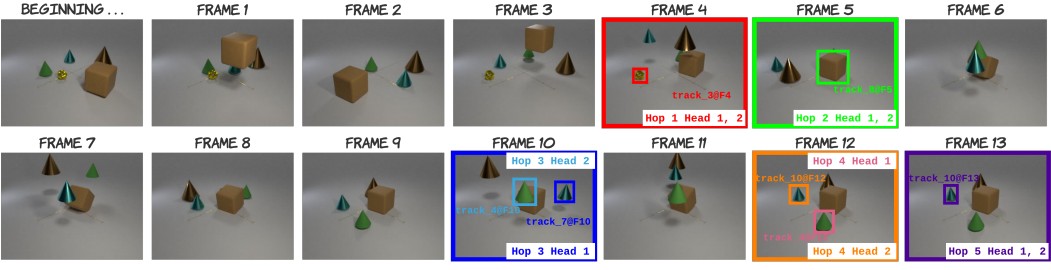

(a) Snitch is contained by the brown cone at frame 5 (the last visible snitch is in frame 4). Brown cone moves to the back of cube at frame 6. After that, ***the snitch is behind of the cube*** till the end.

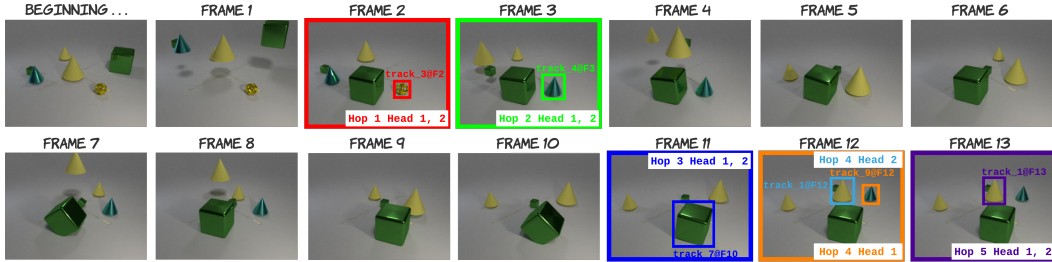

(b) Snitch is contained by the cyan cone at frame 3 (the last visible snitch is in frame 2). After that, cyan cone is contained by the yellow cone twice (frame 5 and frame 9), but in the end the cyan cone is visible and ***the snitch is inside of the cyan cone***.

**Figure 15:** Failure cases. `Hopper` produces wrong Top-1, Top-5 prediction and terrible $L_1$ results for these failure cases. Similarly, we highlight the attended object per hop and per head (**Hop1**, **Hop2**, **Hop3**, **Hop4**, and **Hop5**). **Case (a).** 'CATERh_048295': the occlusion has made the Snitch Localization task extremely difficult since when the snitch got occluded it was contained by the brown cone. Meanwhile, `Hopper` fails to attend to the immediate container of the last visible snitch (should be the brown cone at frame 5) in Hop 2. **Case (b).** 'CATERh_022965': the snitch was not visible very early in the video (at frame 3), the recursive containment, as well as the presence of two similar looking cones have made the task extremely difficult. `Hopper` fails to attend to the correct object in Hop 3 (should be the yellow cone).

## I RELATED WORK (FULL VERSION)

In this section, we provide detailed discussion of related work. Our work is generally related to the following recent research directions.

**Video understanding & analysis.** With the release of large-scale datasets such as Kinetics (Carreira & Zisserman, 2017), Charades (Sigurdsson et al., 2016), and Something something (Goyal et al., 2017), the development of video representation has matured quickly in recent years. Early approaches use deep visual features from 2D ConvNets with LSTMs for temporal aggregation (Donahue et al., 2015; Yue-Hei Ng et al., 2015). As a natural extension to handle the video data, 3D ConvNets were later proposed (Ji et al., 2012; Taylor et al., 2010) but with the issue of inefficiency and huge increase in parameters. Using both RGB and optical flow modalities, Two-stream networks (Simonyan & Zisserman, 2014; Feichtenhofer et al., 2016) and Two-Stream Inflated 3D ConvNets (I3D) (Carreira & Zisserman, 2017) were designed. With the emphasis on capturing the temporal structure of a video, TSN (Wang et al., 2016), TRN (Zhou et al., 2018), TSM (Lin et al., 2019) and TPN (Zhou et al., 2018) were successively proposed and gained considerate improvements. Recently, attention mechanism and Transformer design (Vaswani et al., 2017) have been utilized for more effective and transparent video understanding. Such models include Non-local Neural Networks (NL) (Wang et al., 2018) that capture long-range spacetime dependencies, SINet (Ma et al., 2018) that learns higher-order object interactions and Action Transformer (Girdhar et al., 2019) that learns to attend to relevant regions of the actor and their context. Nevertheless, instead of the reasoning capabilities, existing benchmarks and models for video understanding and analysis mainly have focused on pattern recognition from complex visual and temporal input.

**Visual reasoning from images.** To expand beyond image recognition and classification, research on visual reasoning has been largely focused on Visual Question Answering (VQA). For example, a diagnostic VQA benchmark dataset called CLEVR (Johnson et al., 2017a) was built that reduces

spatial biases and tests a range of visual reasoning abilities. There have been a few visual reasoning models proposed (Santoro et al., 2017; Perez et al., 2017; Mascharka et al., 2018; Suarez et al., 2018; Aditya et al., 2018). For example, inspired by module networks (Andreas et al., 2016b;a), Johnson et al. (2017b) propose a compositional model for visual reasoning on CLEVR that consists of a program generator that constructs an explicit representation of the reasoning process to be performed, and an execution engine that executes the resulting program to produce an answer; both are implemented by neural networks. Evaluated on the CLEVR dataset, Hu et al. (2017) proposed N2NMNs, i.e., End-to-End Module Networks, which learn to reason by directly predicting the network structures while simultaneously learning network parameters. MAC networks (Hudson & Manning, 2018), that approach CLEVR by decomposing the VQA problems into a series of attention-based reasoning steps, were proposed by stringing the MAC cells end-to-end and imposing structural constraints to effectively learn to perform iterative reasoning. Further, datasets for real-world visual reasoning and compositional question answering are released such as GQA (Hudson & Manning, 2019b). Neural State Machine (Hudson & Manning, 2019a) was introduced for real-world VQA that performs sequential reasoning by traversing the nodes over a probabilistic graph which is predicted from the image.

**Video reasoning.** There has been a notable progress for joint video and language reasoning. For example, in order to strengthen the ability to reason about temporal and causal events from videos, the CLEVRER video question answering dataset (Yi et al., 2020) was introduced being a diagnostic dataset generated under the same visual settings as CLEVR but instead for systematic evaluation of video models. Other artificial video question answering datasets include COG (Yang et al., 2018a) and MarioQA (Mun et al., 2017). There have been also numerous datasets that are based on real-world videos and human-generated questions such as MovieQA (Tapaswi et al., 2016), TGIF-QA (Jang et al., 2017), TVQA (Lei et al., 2018) and Social-IQ (Zadeh et al., 2019). Moving beyond the question and answering task, CoPhy (Baradel et al., 2019) studies physical dynamics prediction in a counterfactual setting and a small-sized causality video dataset (Fire & Zhu, 2017) was released to study the causal relationships between human actions and hidden statuses. To date, research on the general video understanding and reasoning is still limited. Focusing on both video reasoning and a general video recognition and understanding, we experimented on the recently released CATER dataset (Girdhar & Ramanan, 2020), a synthetic video recognition dataset which is also built upon CLEVR focuses on spatial and temporal reasoning as well as localizing particular object of interest. There also has been significant research in object tracking, often with an emphasis on occlusions with the goal of providing object permanence (Bewley et al., 2016; Wojke et al., 2017; Li et al., 2018; Zhu et al., 2018; Wang et al., 2019b). Traditional object tracking approaches have focused on the fine-grained temporal and spatial understanding and often require expensive supervision of location of the objects in every frame (Shamsian et al., 2020). We address object permanence and video recognition on CATER with a model that performs tracking-integrated object-centric reasoning for localizing object of interest.

**Multi-hop reasoning.** Reasoning systems vary in expressive power and predictive abilities, which include systems focus on symbolic reasoning (e.g., with first order logic), probabilistic reasoning, causal reasoning, etc (Bottou, 2014). Among them, multi-hop reasoning is the ability to reason with information collected from multiple passages to derive the answer (Wang et al., 2019a). Because of the desire for chains of reasoning, several multi-hop datasets and models have been proposed for nature language processing tasks (Dhingra et al., 2020; Dua et al., 2019; Welbl et al., 2018; Talmor & Berant, 2018; Yang et al., 2018b). For example, Das et al. (2016) introduced a recurrent neural network model which allows chains of reasoning over entities, relations, and text. Chen et al. (2019) proposed a two-stage model that identifies intermediate discrete reasoning chains over the text via an extractor model and then separately determines the answer through a BERT-based answer module. Wang et al. (2019a) investigated that whether providing the full reasoning chain of multiple passages, instead of just one final passage where the answer appears, could improve the performance of the existing models. Their results demonstrate the existence of the potential improvement using explicit multi-hop reasoning. Multi-hop reasoning gives us a discrete intermediate output of the reasoning process, which can help gauge model's behavior beyond just final task accuracy (Chen et al., 2019). Favoring the benefits that multi-hop reasoning could bring, in this paper, we developed a video dataset that explicitly requires aggregating clues from different spatiotemporal parts of the video and a multi-hop model that automatically extracts a step-by-step reasoning chain. Our proposed Multi-hop Transformer improves interpretability and imitates a natural way of thinking.

The iterative attention-based neural reasoning (Locatello et al., 2020; Hudson & Manning, 2019a) with a contrastive debias loss further offers robustness and generalization.

## J  MHT ARCHITECTURE

We illustrate the architecture of the proposed Multi-hop Transformer (MHT) in Figure 16.

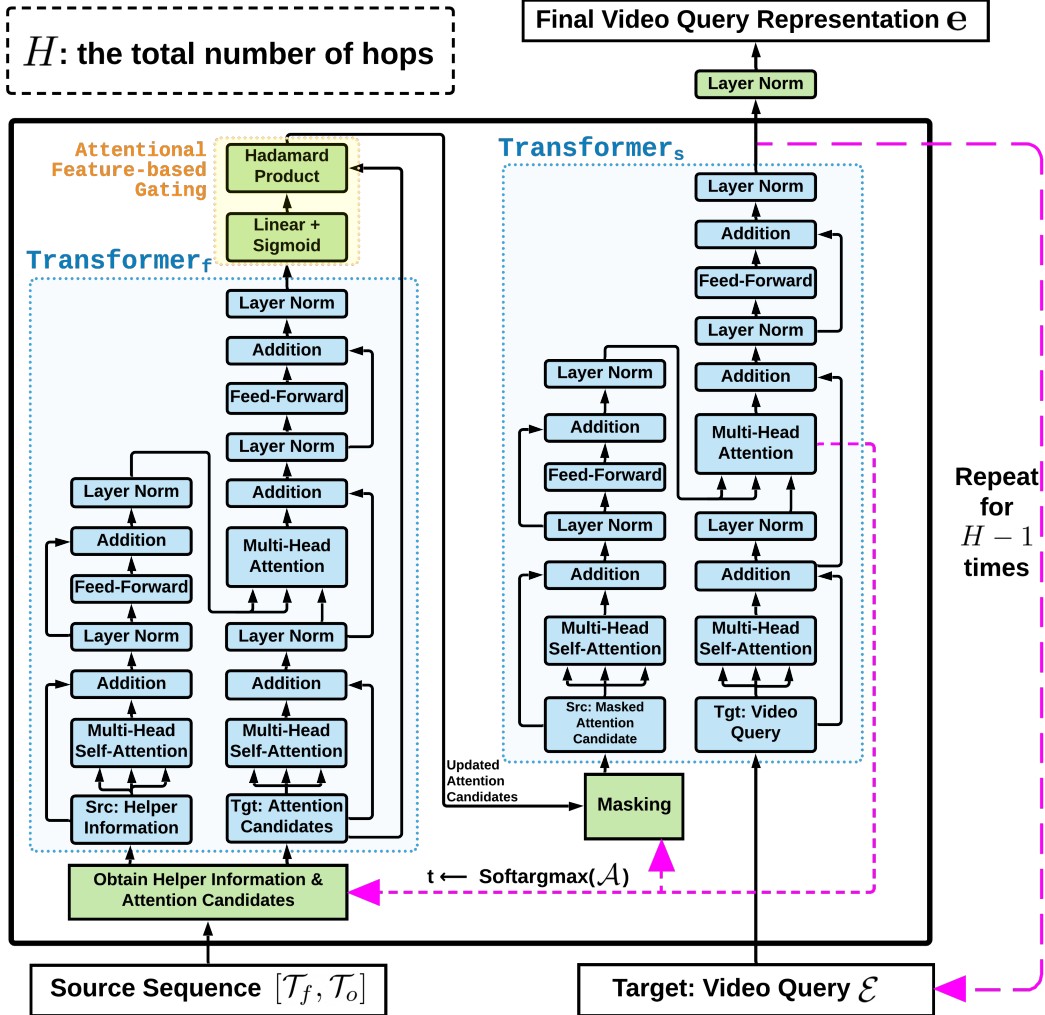

**Figure 16: Architecture of the Multi-hop Transformer (MHT) that learns a comprehensive video query representation and meanwhile encourages *multi-step compositional long-term reasoning* of a spatiotemporal sequence.** As inputs to this module, the 'Source Sequence' is $[\mathcal{T}_f, \mathcal{T}_o]$, where [ , ] denote concatenation; and the 'Target Video Query' is $\mathcal{E} \in \mathbb{R}^{1 \times d}$. 'Final Video Query Representation' is $\mathbf{e} \in \mathbb{R}^{1 \times d}$. $\mathcal{A} \in \mathbb{R}^{NT \times 1}$ refers to attention weights from the encoder-decoder multi-head attention layer in Transformer$_s$, averaged over all heads. To connect this figure with Algorithm 1, 'Obtain Helper Information & Attention Candidates' refers to line 5 for 'Helper Information' $\mathcal{H}$ (or line 7 for the first iteration), and line 9 for the 'Attention Candidates' $\mathcal{U} \in \mathbb{R}^{NT \times d}$. Dimensionality of 'Helper Information' $\mathcal{H}$ is $T \times d$ for hop 1 and $N \times d$ for the rest of the hops. Transformer$_f$ and Transformer$_s$ are using the original Transformer architecture (Vaswani et al., 2017). 'Attentional Feature-based Gating' corresponds to line 11. 'Updated Attention Candidates' is $\mathcal{U}_{\text{update}} \in \mathbb{R}^{NT \times d}$. 'Masking' corresponds to line 12. Its output, 'Masked Attention Candidates' is $\mathcal{U}_{\text{mask}} \in \mathbb{R}^{NT \times d}$ (however, certain tokens out of these $NT$ tokens are masked and will have 0 attention weights). The last 'Layer Norm' after the Transformer$_s$ corresponds to line 16. $H$ denotes the total number of hops, i.e., total number of iterations, and it varies across videos. Please refer to Section 4 for details.

