# OpenReview forum: "Hopper: Multi-hop Transformer for Spatiotemporal Reasoning"
_ICLR.cc/2021/Conference — ICLR 2021 Poster_

### Official Review · AnonReviewer1 · 2020-10-22
**Recommend accept**

**Rating:** 8
**Confidence:** 4

**Review:**

This paper presents Hopper, a method that performs multi-hop reasoning to address the problem of object permanence in videos.

### Strengths:
- Very good results in the Snitch Localization task. Good and sensible baselines.
- Good ablations, experiments and visualizations. Also good justification of results both for Hopper and for baselines.
- They introduce a method that is modular, and clearly separates several stages of processing (while still making everything end-to-end trainable). This provides good understanding of the system, and makes ablations and comparisons with baselines easier (for example, baselines can work a the per-module level).
- The multi-hop transformer reasoning module they introduce is intuitive, and it is very well executed.
- Overall well written and good positioning among state of the art literature.
- Contribution of a new CATER-h dataset that corrects some biases of CATER for the specific task the paper is solving. Good explanation of the need for this dataset, both by analyzing the dataset statistics, and by looking at the results of the baselines on the two datasets.

### Weaknesses:
- A lot of supervision and limited dataset.
	- Object annotation class and bounding boxes are available.
	- Fixed set of objects and attributes, with no variation across samples (synthetic data).
- The tracker learns based on the class. If there are two objects belonging to the same class, it will probably not distinguish them (see Figure 15a). Probably humans could watch a video and follow the snitch even if all objects looked exactly the same, just by following their movements, and this method is, _by construction_, not doing this. Therefore, Figure 15a is not just some very hard case, but a case that the system as it stands cannot solve. This implies that the tracking done by the system is a very weak form of tracking, and it is more a relabeling of objects. Actually, to solve the task the system only needs to know 1) where the snitch was before and 2) where did the container move (just reidentify by classification). While this can be improved, the main contribution of the paper is in the step after this, so it is OK to leave it for future work.
- In some cases the method feels very specific for the presented task and dataset. This is acceptable to some extent because it is a hard problem and it doesn't have to scale right away, but I am concerned it is too specific about this formulation and cannot be used. There are a lot of specific intuitions for this specific problem and setting (not interesting for any other case). See for example page 6. More specifically, an example is the heuristic for computing the occluder, which would fail if the snitch has moved between frames, and it is visible but somewhere else (and this case is not even generalizing to a slightly different task, but a potential situation in the current dataset). While these heuristics are optional and the authors present ablations, the feeling is general for the whole method.
- Some parts of the method could use more clarity:
	- Figure 3 is confusing, as it does not exactly follow Algorithm 1. It would be convenient to use the same names as in the rest of the paper for the layers and the inputs/outputs.
	- Why the attentional feature-based gating (line 11 algorithm 1) is necessary? Why doesn't Transformer_f directly output the final U_update?
	- Why two transformers are necessary, actually? The masking could be applied at the beginning of the first one and just use one. Note this is not the same as the "hopper-transformer" in the baselines.
	- t is computed with a softargmax. However, it is used in places that require it being a hard number, for example in line 2 in the algorithm. This implies these places have to detach the gradient from t. Apart from line 2, is there any other place that requires this? What about lines 12 (Masking) and 5 (Extract). My understanding of the paper tells me that at least the Masking() module requires a hard t. In what case is the gradient actually propagated through t?
	- Teacher forcing: the paper argues that hop 2 should focus on the first occluder. What about hop 3, what is it supposed to predict?
	- For line 14 in the algorithm, are the attentions of the heads averaged?
- About the 5 steps: Looking at the examples (eg Figure 15b) it looks like there are much more than just 5 key steps to follow. However, Hopper never uses H > 5 in the shown examples (while H=5 is actually the minimum number of hops that are allowed to the system). Why is that? It looks like as soon as it has a chance, the system tries to predict the last frame available, even if it is not the most convenient (it can always get later, to the last frame). This could make sense for CATER, but the shown examples are for CATERh . This is important because the main point of Hopper is that it can select where to attend. I would appreciate some intuitive explanation and statistics of the number of steps Hopper takes.
- Unclear what the model is _really_ learning. While it is not learning temporal biases, there are potentially a lot of other biases it can be exploiting. It would be very interesting to have analyses to answer this question. Otherwise it is hard to believe the models are really "solving" this task.

### Additional comments and questions:
- When an object is occluded or contained by another one, the goal is to predict the position of the first object, or the position of the occluder?
- Is the 1 fps chosen for any specific reason? The solution to the problem is actually different depending on the sampling rate. For example, it would be possible that in between frames there is a key move that we do not see.
- "Humans realize object permanence by identifying key or critical frames where objects become hidden", and similar sentences throughout the paper. Citation?
- I think the third sentence in the introduction refers to the first one. It is confusing because it reads as it is referring to the second one instead. I would move the second one to later when the authors talk about object permanence (line 20).
- If the key point of the paper is the multi-hop transformer, an interesting comparison would be the same system without DETR, and instead using only the true class label and bounding box (remove steps A and B). At least to see how much these are a bottleneck, or how important they really are.
- Analyses on learned attributes. The classes are combinations of 4 attributes. Why not learning them separately? Is the model creating different represenations for each specific combination? Would it generalize to a new object or to a new combination of attributes not seen during training?
- Will the code for the paper and the cater-h dataset be released?

### Final recommendation
Overall, I believe this paper should be clearly accepted to ICLR, as the strengths outweigh the weaknesses. This paper is not the final solution to the object permanence task, and it has a lot of possible improvements, but it is a good step.

---

> ### Author Response · Authors · 2020-11-20
> **Thanks for the positive review! We answer your questions below. - Part 1**
>
> Thank you for your positive review and helpful notes! We have improved our manuscript based on your feedback. Please see the following for our response point-by-point.
>
> **1. Regarding “a lot of supervision and limited dataset.”:
> ===>**
>
> In our paper, the object-based supervision is only used for training the object detector, and compared to the existing work, we use much less supervision. For example, our reasoning module (Multi-hop Transformer) does not use any object-based annotations, or any task-oriented per-frame ground-truth supervision. With respect to the limitation on the dataset, even though we agree the current synthetic dataset is limited, the CATER dataset is an excellent benchmark given that the task is far from being addressed.
>
> **2. Regarding “the tracking done by the system is a very weak form of tracking”:
> ===>**
>
> We agree with your assessment that our tracking method is not state of the art and the major contribution of the paper is the reasoning module. We’ve deemphasized the importance of our tracking method in the paper and added clarification on this at the end of Section 3.3 (Tracking).
>
> **3. Regarding “In some cases the method feels very specific for the presented task and dataset”:
> ===>**
>
> Regarding these training techniques, even though some of them are sort of specific to the Snitch Localization task, we believe they suggest some interesting insights. For example, the object & frame auxiliary losses for hop 1 and hop 2 (which are specific to Snitch Localization) indicate that providing supervision on early hops are crucial towards learning a high accuracy multi-hop model. The other proposed techniques are general and applicable to other datasets and tasks. The “dynamic hop” and “minimal hop constraint” are methods that could be inspirational to other multi-hop reasoning related research. Teacher forcing and the contrastive debiasing loss are inspired from existing research, and are general for many models, tasks, and datasets (e.g., real-world action recognition datasets). Furthermore, our proposed Hopper framework is modular and general, as you have pointed out.
>
> **4. Regarding “Figure 3 is confusing”:
> ===>**
>
> We’ve adopted your suggestion, and improved the figure of the Multi-hop Transformer. We also added a more detailed caption to connect the figure with Algorithm 1.
>
> **5. Regarding “Why the attentional feature-based gating (line 11 algorithm 1) is necessary?”:
> ===>**
>
> We agree that the description on motivation of this layer is lacking. We’ve added clarification on this in Section 4 of the paper, and an ablation experiment by removing the Attentional Feature-based Gating layer (line 11 in Algorithm 1) to empirically show its benefits (Top-1 66.6%, Top-5 87.4%, and L1 1.16; Results are in Table 4 of the manuscript). In a nutshell, the Attentional Feature-based Gating layer is to combine the Transformer_f modified representation with the original representation of $\mathcal{U}$. This layer, added on top of Transformer_f, *provides additional new information*, because it switches the perspective into learning new representations of objects *by learning a feature mask to select salient dimensions of $\mathcal{U}$*, conditioned on $\mathcal{U}_{update}$ produced by Transformer_f, with the Sigmoid gate mechanism.
>
> **6. Regarding “Why are two transformers necessary? The masking could be applied at the beginning of the first one and just use one.”:
> ===>**
>
> We’ve improved Section 4 to address this question. At a high level, at each reasoning hop the model should attend to one or a few objects in a frame. At every hop the model re-evaluates all the attention candidates (object representations) in the context of the previous reasoning step, then selects a new subset of objects to attend to and uses them to update the information in the query encoding. At the end of the reasoning chain, the query encoding should contain the information required to provide the final answer.
>
> Transformer_f performs the re-evaluation of the attention candidates by utilizing the information related to the previous reasoning step (or coarse-grained global image context for hop 1) and adapting the representations of object entities. Transformer_s learns the latent representation of the video query by narrowing down the set of objects to attend to and attentively utilizing the information extracted from Transformer_f (along with the Attentional Feature-based Gating layer if to be more accurate).
>
> We’ve added a result of an ablation experiment by feeding in masked object representations for both Transformer_f  and Transformer_s (Top-1 65.7%, Top-5 88.4%, and L1 1.13, in Table 4). This experiment shows the benefits of applying masking only to the second transformer.
>
> **7. Regarding “t is computed with a softargmax. However, it is used in places that require it being a hard number.”:
> ===>**
>
> Your understanding is correct, regarding line 2, 12 and 5. “Detach()” is happening for these three places.

---

> > ### Author Response · Authors · 2020-11-20
> > **Thanks for the positive review! We answer your questions below. - Part 2**
> >
> > **8. Regarding “what about hop 3, what is it supposed to predict”:
> > ===>**
> >
> > We only design heuristic-based weak supervision for hop 1 and 2, this is because for the Snitch Localization task, depending on the specific case of a video, whom to attend to in hop 3 should be different. If the model attends to a container (e.g., a cone that covers snitch) in hop 2, whom to attend to in hop 3 could be a container of the previous container, or new immediate container of snitch (indicating the previous cone has "un-contained" the snitch), etc., in any future frames.  Such diverse possibility brings unnecessary implementation challenges, and it is always more tempting if the model could automatically determine this without supervision.
> >
> > **9. Regarding “for line 14 in the algorithm, are the attentions of the heads averaged?”:
> > ===>**
> >
> > Yes, attention weights are averaged from all heads. We have clarified this now in the manuscript.
> >
> > **10. Regarding “the number of steps Hopper takes”:
> > ===>**
> >
> > We’ve added a new section in the Appendix to analyze the number of steps that Hopper takes. Hopper uses $H$ > 5 for several times, but it indeed has the tendency to attend to later frames. We believe it is because no supervision is provided for the intermediate hops.  As the final Snitch Localization task itself is largely focused on the last frame of the video, without supervision for the intermediate hops, the model tends to “look at” later frames as soon as possible. These results suggest where we can improve for the current model, e.g., one possibility is to design self-supervision for each intermediate hop. Please refer to Appendix C.1 for more details.
> >
> > **11. Regarding “there are potentially a lot of other biases”:
> > ===>**
> >
> > It is true that there can be potentially other dataset biases. One bias we observed is that the snitch can only be contained by cones. Most methods learn to predict the location of one of the visible cones in the last frame when the snitch is not present. This is due to the nature of objects used in CATER and the fact that only the cone can carry the snitch. Furthermore, because of the animation rules of CATER, there might exist other dataset biases, such as bias in terms of object size and shape, etc., which are hard to discover and address. This highlights the challenge in building a fully unbiased (synthetic or real) dataset. CATER-h addresses the temporal bias that CATER has. In addition, to have a high accuracy on CATER-h, a model needs to perform spatiotemporal reasoning to locate the *correct* cone that covers snitch (instead of a random visible cone). We now describe this in Appendix G.6.
> >
> > **12. Regarding “when an object is occluded or contained by another one, the goal is to predict the position of the first object, or the position of the occluder?”:
> > ===>**
> >
> > The goal is to predict the position of the first object.
> >
> > **13. Regarding “is the 1 fps chosen for any specific reason?”:
> > ===>**
> >
> > Since the task is to determine where the snitch is in the end of the video, and we observe that the motions are not rapid, we use 1 FPS for system optimization reasons (e.g., GPU memory constraints). Lower FPS implies less computation so it is usually more favorable.
> >
> > **14. Regarding missing citation:
> > ===>**
> >
> > We’ve addressed this issue and added citation (Bremner et al., 2015).
> > .
> >
> > **15. Regarding “the third sentence in the introduction refers to the first one. It is confusing because it reads as it is referring to the second one instead. I would move the second one to later when the authors talk about object permanence (line 20)”:
> > ===>**
> >
> > Thanks for the note! Based on your suggestion, we’ve improved the first paragraph of the Introduction section.
> >
> > **16. Regarding “an interesting comparison would be the same system without DETR, and instead using only the true class label and bounding box”:
> > ===>**
> >
> > We do find this idea interesting, which we might consider in our future work. Currently, we are not able to do so because CATER or CATER-h does not have the ground truth class label and bounding box for every object in every frame.

---

> > > ### Author Response · Authors · 2020-11-20
> > > **Thanks for the positive review! We answer your questions below. - Part 3**
> > >
> > > **17. Regarding “why not learn object attributes separately? Is the model creating different representations for each specific combination? Would it generalize to a new object or to a new combination of attributes not seen during training?”:
> > > ===>**
> > >
> > > Currently, the object detector is creating different representations for each specific combination, because of this, we think it cannot generalize to new combinations of attributes. The object detector is not the focus of our contribution, and any other detector could be easily integrated into the framework.
> > >
> > > We did not learn object attributes separately, because 1) It is more universal for object detectors to predict just the object class label and the bounding box; 2) If we form our Hopper framework in this way, Hopper would be less general, because it indicates more expensive labeling when applied to other datasets (or tasks), and object detection dataset with such feature is much fewer; 3) The current object class annotation is offered in an attributes-combined way; 4) Learning object attributes separately might be an easier task (and might even reduce the error rate or the number labeled images needed to learn). It would be easy to modify the detector to predict individual attributes rather than the combination, which we believe would have the capability to generalize to new combinations.
> > >
> > >
> > > **18. Regarding “will the code for the paper and the cater-h dataset be released?”:
> > > ===>**
> > >
> > > We will release the CATER-h dataset as well as additional code and documentation.

---

### Official Review · AnonReviewer4 · 2020-10-26
**Major concerns about the algorithm description.**

**Rating:** 6
**Confidence:** 4

**Review:**

This paper introduces an architecture (Multi-Hop Transformer) for spatio-temporal reasoning in video, focusing on a localisation task for scenes where the object of interest is often occluded (Snitch Localisation task in CATER). The model extracts objects using an external object-detector and predicts objects’ trajectories using the Hungarian algorithm. The Multi-Hop Transformers learns to hop over unnecessary frames, by focusing only on a set of few critical steps (at each time step, the next critical frame is the one containing the most attended object). To alleviate the problems encountered during the training procedure (such as error propagation), several auxiliary training methods are proposed to guide the first few hops or to ensure contrastive debias. Additionally, a harder version of the existing CATER dataset is created, to alleviate the temporal bias existent in the previous version of the dataset.

Pro:
- The “dynamic stride” proposed in the Multi-Hop Transformer is a novel and interesting idea, showing promising results in tasks requiring object permanence. Moreover, the combination of global features and object-centric representations is sound and suitable.
- The ablation studies (Table 1-2 but also those from the supplementary material) are carefully designed, showing the importance of each component and offering a fair comparison between the methods.
- The temporarily unbiased version of CATER dataset, where the last visible snitch is distributed uniformly across the length of the video, is crucial for improving the evaluation of different models in video reasoning tasks.

Cons:
The method is poorly described in the paper (especially in Section 4): the writing is very hard to follow, without a clear structure of ideas. Several details about the sub-modules are missing and the proposed techniques are not supported by any intuitive motivation.  Some example for this are the followings:
- In the second layer, the Transformer_f module receives as input both U and H. Does it send messages across all types of edges (between tokens in U, between tokens in H and across them)? Same question for Transformer_s.
- Since the video query should be a single token (the input has dimension 1xd, thus making me think that it is preserved along the algorithm since additional information about the intermediary shapes are not provided), what is the purpose of the first self-attention layer (column 4 in Fig. 3). What tokens should this layer combine?
The meanings of each latent representation are not provided (or presented too vaguely e.g. “by reasoning about the relations between the object entities and how would each object entity relate to the reasoning performed by the previous hop or global information”). From the paper, it seems that the \epsilon variable starts by representing a query embedding, but becomes a video representation immediately after. Should that token be seen as an encoding for the entire video, or as a representation of the object of interest across the video (the snitch in this case)?
- Why does the Transformer_f send messages between all time steps and not only the U_masked?
- Since CLEVR_h has a uniform distribution of the last visible frame, why does the Hopper-transformer (last frame) have such a high performance (in Table 2)? Is it able to learn anything else besides the 7% cases when the snitch becomes last visible in the 13th frame?
- The purpose of the tracking module (the Hungarian algorithm applied on top of predicted objects) is not clearly mentioned. Is it used only for consistency reasons,  to help the model in preserving the positional index across frames, or do those matching-edges guide the transformers in any way?
- The proposed architecture is composed of several big components (transformers). Thus a comparison against existing methods in terms of the number of parameters and flops would be useful.
- Since the additional training techniques bring considerable improvement, it seems to me that the proposed algorithm is not very robust to different kinds of noise encountered during training. This could be a problem when applying to real-world scenarios when probably the object detector will not be as accurate as the one used in this synthetic setup.  Do you think that the general algorithm, but especially all the heuristics proposed for the auxiliary losses, will suffer because of that?

Minor:
- In G.1 (appendix) there is a typo: “stacking 6 transformer decoder layers and 6 transformer decoder layers”
- In the “Interpretability” paragraph, the sentence … learns to perform snitch-oriented tracking automatically, without having per-frame supervision“ is confusing. Even if no snitch-level supervision is used to guide the localisation across video, the per-frame supervision is used to learn the detector. I agree that other datasets could be used in real-world scenarios, but in this case, the per-frame annotation is used instead.

I really think that the idea of multi-hop and “selective” pointers in transformers is valuable and useful for video reasoning. Moreover, I appreciate the empirical study presented in this work. However, I have major concerns about the way the algorithm is explained in the paper. Several crucial parts are not clearly explained and the usage of the two transformers inside the multi-hop method are not motivated at all. Taking these into consideration, I don’t think that this work is in a proper form to be published since several sections should be rewritten entirely. In conclusion, I recommend the rejection.

########### UPDATE #########

I thank the authors for clarifying the method description, providing additional experiments and actively engaging in discussion. Since my main concern regarding the writing part was addressed I change my rating and I agree with the acceptance.

---

> ### Author Response · Authors · 2020-11-20
> **Algorithm description (Section 4) has been revised - Part 1**
>
> Thank you for the constructive feedback! We genuinely appreciate that you find our ideas "novel and interesting", "sound and suitable", ablation studies are "carefully designed" and comparisons are "fair". Based upon your suggestions, we have significantly revised the paper, especially Section 4 (Multi-hop Transformer) to address your primary concern about the algorithm description. We've added the motivations of the sub-modules, as well as more detailed and structured description of them. In the following, we answer each question point by point (as well as adding them in the paper as appropriate).
>
> **1. Regarding “Transformer_f receives as input both U and H. Does it send messages across all types of edges (between tokens in U, between tokens in H and across them)? Same question for Transformer_s”:
> ===>**
>
> Message passing in Transformer_f is performed across all connections between tokens in $\mathcal{U}$, between tokens in $\mathcal{H}$, and especially across $\mathcal{U}$ and $\mathcal{H}$. On the other hand, message passing in Transformer_s is only performed between tokens in $\mathcal{E}$ (which has only 1 token for Snitch Localization), between tokens in *unmasked* tokens in  $\mathcal{U}_{update}$, and more importantly, across connections between the video query $\mathcal{E}$ and *unmasked* tokens in $\mathcal{U}${update}. We’ve added a description of this in the “Transformer_s” subsection.
>
> **2. Regarding “what is the purpose of the first self-attention layer (column 4 in Fig. 3). What tokens should this layer combine?”:
> ===>**
>
> You are correct in pointing out that the first multi-head self-attention layer in the decoder part of Transformer_s does not have any attention effect, since the video query is indeed just one token (attention weights are just always 1s). The layer was placed to depict multi-query inputs but for CATER (and CATER-h) this can be removed. We keep it to make the model more general and simpler (easier for the reader to understand). We ran experiments and removing that self-attention layer in Transformer_s does not significantly change the results (Top-1 68.2%, Top-5 88.2%, and L1 1.13). We have added an explanation in the paper to avoid confusion in the “Summary & discussion” subsection of Section 4.
>
> **3. Regarding “the \epsilon variable starts by representing a query embedding, but becomes a video representation immediately after. Should that token be seen as an encoding for the entire video, or as a representation of the object of interest across the video (the snitch in this case)?”:
> ===>**
>
> $\mathcal{E}$  should be seen as an encoding for a query of the entire video (i.e., snitch localization is a query, but sphere localization could be a query and in other tasks, action classification could also be a query). We’ve added this explanation in the “Summary & discussion” subsection of Section 4, and made it consistent across the paper.
>
> **4. Regarding “why does the Transformer_f send messages between all time steps and not only the U_masked?”:
> ===>**
>
> Thanks for asking this question. We’ve added explanations on this in Section 4. This is because, 1) Transformer_f is designed to gather information for Transformer_s, and Transformer_s will narrow down to an “answer” (object of a frame it should mostly attend to in this hop, under the context of an object permanence task) from a candidate “answer” pool, so only Transformer_s uses  $\mathcal{U}_{masked}$ (the actual candidate “answer” pool) to implement the autoregressive processing. 2) Object representations are not perfect due to errors occasionally made by the object detector, hence, global information such as image representations are useful sometimes. In addition, to determine which object a model should attend to for frames from t+1 to END, objects in frame t (or even earlier) might also be beneficial. This is why we have the concept of helper information $\mathcal{H}$ (i.e, extra useful information outside of the candidate “answer” pool). Thus, Transformer_f is designed to condense useful information, and it adapts object representations by attending to objects in all time steps (via self-attention) and helper information $\mathcal{H}$ (via encoder-decoder attention). Then, adapted object representations will be masked for Transformer_s to narrow down to an “answer”.
>
> In addition, we’ve implemented a version that the target input for Transformer_f is also *masked*  $\mathcal{U}$. We added the results of this version in Table 4 (Top-1 65.7%, Top-5 88.4%, and L1 1.13). We’ve empirically verified that it is beneficial for the Transformer_f to use just  $\mathcal{U}$.

---

> > ### Author Response · Authors · 2020-11-20
> > **Algorithm description (Section 4) has been revised - Part 2**
> >
> > **5. Regarding “why does the Hopper-transformer (last frame) have such a high performance (in Table 2)?”:
> > ===>**
> >
> > We also hypothesized that the reason it has 41.8% Top-1 accuracy on CATER-h might be the existence of other dataset biases (apart from the temporal bias that CATER has). Upon further investigation, we identify one type of additional bias, the “cone bias”, i.e., snitch can only be contained by a cone in the videos of CATER and CATER-h.  In order to verify the existence of the “cone bias”, we compute the accuracy if we make a random guess among the grids of cones that are not covered by any other cones, for all test videos whose snitch is covered in the end. This gives us 48.26% Top-1 accuracy. This shows that the “cone bias” does exist in the CATER-h dataset (it also exists in CATER).
> >
> > However, the “cone bias” is due to the nature of objects used in CATER, and is rather more close to a feature of the dataset. Because of the animation rules of CATER, there might exist other potential dataset biases, such as object bias in terms of object size and shape, etc., which are hard to discover and address. This highlights the glaring challenge in building a fully unbiased (synthetic or real) dataset. CATER-h addresses the temporal bias that CATER has, and correcting this temporal bias is the most important for a video dataset. In addition, to have a high accuracy on CATER-h, a model needs to perform spatiotemporal reasoning to locate the *correct* cone that covers snitch (instead of a random visible cone).
> >
> > We believe this finding is also important and interesting for the research community. We've added discussion on this in Appendix G.6 (also mentioned in the “Results” subsection of the main paper).
> >
> > **6. Regarding “the purpose of the tracking module”:
> > ===>**
> >
> > The tracking module is used for consistency reasons. The motivation of applying a tracking module beforehand is that tracking produces consistent object representations as it links the representations of each object through time. This more consistent object representation could make the learning of Multi-hop Transformer easier. To demonstrate the role of the tracking module, we added an ablation and now show Hopper-multihop results of *not* using tracking in Table 4, which is worse (Top-1 67.5%, Top-5 88.3%, and L1 1.14). The motivation of using the tracking module has also been added to the beginning of Section 3.3.
> >
> > **7. Regarding “comparison against existing methods in terms of the number of parameters and flops”:
> > ===>**
> >
> > We've added Table 5 and compared the number of parameters of our Hopper-multihop with alternative methods. Our proposed method is the most efficient one in terms of the number of parameters. This is because of the iterative design in our Multi-hop Transformer that allows for parameter sharing across iterations. Please see Appendix B for more details. We will report statistics on the number of FLOPs in our camera-ready version of the paper.
> >
> > **8.  Regarding “the additional training techniques bring considerable improvement” and “when applying to real-world scenarios, probably the object detector will not be as accurate as the one used in this synthetic setup. Do you think that the general algorithm, but especially all the heuristics proposed for the auxiliary losses, will suffer because of that?”:
> > ===>**
> >
> > The training techniques are part of our approach, and suggest some interesting insights (e.g., providing supervision on early hops and teacher forcing are beneficial for training a multi-hop model, the contrastive loss is good for debiasing and model interpretability). In addition, these heuristic-based loss terms are used to mitigate the issues of having no ground truth reasoning chain. Furthermore, some of the training techniques, such as the contrastive debiasing loss, offer more robustness to the model.
> >
> > In terms of whether these heuristics proposed for the auxiliary losses suffer from a less good object detector, yes, we think so. The heuristic-based loss for intermediate hops will be less accurate. We added a “Summary & discussion” subsection for both Section 4 and Section 5 with related contents to address your concerns.
> >
> > **9. Regarding the typo “stacking 6 transformer decoder layers and 6 transformer decoder layers”:**
> > ===>
> >
> > The typo has been corrected. Thank you for your careful reading!
> >
> > **10. Regarding our claim on “without having per-frame supervision”:
> > ===>**
> >
> > We’ve acknowledged your concerns and removed that sentence from the paper.

---

> > > ### Author Response · Authors · 2020-11-20
> > > **Algorithm description (Section 4) has been revised - Part 3**
> > >
> > > **11. Regarding the motivation of usage of the two transformers inside the Multi-Hop Transformer:
> > > ===>**
> > >
> > > We’ve improved Section 4 to address this issue. This is inspired by study in cognitive science that human reasoning consists of 2 stages: first, one has to establish the domain about which one reasons and its properties, and only after this initial step can one’s reasoning happen (Stenning & Van Lambalgen, 2012). We first use Transformer_f to adapt the representations of the object entities, which form the main ingredients of the domain under the context of an object-centric video task. Then, Transformer_s is used to produce the task-oriented representation to perform reasoning (toward answering questions about a video).
> > >
> > > At a high level, at each reasoning hop, the Multi-hop Transformer should attend to one or a few objects in a frame. At every hop the model re-evaluates all the attention candidates (object representations) in the context of the previous reasoning step, then selects a new subset of objects to attend to and uses them to update the information in the query encoding. At the end of the reasoning chain, the query encoding should contain the information required to provide the final answer. Transformer_f performs the re-evaluation of the attention candidates by utilizing the information related to the previous reasoning step (or coarse-grained global image context for hop 1) and adapting the representations of object entities. Transformer_s learns the latent representation of the video query by narrowing down the set of objects to attend to and attentively utilizing the information extracted from Transformer_f (along with the Attentional Feature-based Gating layer if to be more accurate).
> > >
> > > We’ve added descriptions of motivations on several design choices including the Attentional Feature-based Gating in the new manuscript. We sincerely hope that the current manuscript is clear now.

---

> > > > ### Comment · AnonReviewer4 · 2020-11-21
> > > > **Re: Algorithm description (Section 4) has been revised**
> > > >
> > > > Thank you so much for the response and for the intense involvement in updating the manuscript to incorporate all the feedback. To me, the model description (for both Hopper and MHT) is much clearer in the current form and the additional experiments reduce my concern regarding the architecture. The “cone bias”  is really interesting, opening other opportunities to increase the difficulty of the task and resulting in better evaluations. Perhaps a dataset where all the objects (except for the snitch) are cones (of different colours but with the same shape and size) would further reduce the biases.  However, the cone finding correlates well with the result of the transformer model reported in the paper.
> > > >
> > > > I will include some additional observations regarding the current form, in case the authors will have the necessary time to address:
> > > >
> > > > - Since Figure 16 was moved in the appendix, the dimensions of different components are missing from the main paper (e.g. U, U_mask, A..). Including them in Section 4 would be helpful.
> > > > - The font used in the main figure (Fig. 2) to indicate the sizes is hard to read. Moreover, including the letters associated with the components (when this is possible) would help to create the connection between the algorithm and the figure. (e.g. T_f, T_o, o, i, e, \epsilon).
> > > > - The function Extract() in line 5 of the algorithm should be referred to in text when that step is presented.
> > > > - Table 4 in Supplementary Material contains some unnecessary columns (dataset and fps).

---

> > > > > ### Author Response · Authors · 2020-11-22
> > > > > **Suggestions are incorporated!**
> > > > >
> > > > > We agree and definitely hope that our findings could lead to additional research in this area towards a dataset with reduced object biases for long-term spatiotemporal understanding. We’ve incorporated all of the new suggestions, i.e., adding dimensionality in Section 4, improving Figure 2, referring to line 5 in the text when it is described, and removing unnecessary columns in Table 4. Thanks!

---

### Official Review · AnonReviewer3 · 2020-10-29
**The paper is an interesting work but more ablation and justification are needed**

**Rating:** 7
**Confidence:** 4

**Review:**


The authors proposed a multi-hop transformer, which takes information encoded in forms of object track and image track as input, to reason over the critical frame sequence to locate the final location of the object of interest. Although I like the idea presented in the paper, I think there are several aspects that will strengthen the paper.
Pros:
1.	The idea of using transformer in a recurrent manner for reasoning in videos is intuitive and interesting.
2.	The authors show the effectiveness of the proposed model by superior quantitative results and the visualization of the most attended object in the inference process of one video sequence.
3.	Thorough ablation is provided for the proposed multi-hop transformer.
4.	The paper is written and presented well.

Cons:
1.	One baseline comparison is missing. The tracking baseline seems to be from prior works rather than from the tracking results produced in the proposed frameworks. It is important to know how well the tracking component itself performs.
2.	More ablation on the framework should be provided. The authors used the DETR for object detection. How does the final performance benefit from using this transformer-based object detection model? Is the proposed framework only compatible with transformer-based object detection?
3.	Although I like the visualization results, it is still a question whether the most attended object is the most important one in the final results or not [1]. It will be interesting to see the gradient visualization rather than the attention visualization.
[1] Sofia Serrano, and Noah A. Smith. 2019. Is Attention Interpretable.arXiv preprint arXiv:1906.03731.



==============


I have read the authors' rebuttal information. The authors have addressed my concerns with additional ablation studies and experiments to verify the effectiveness of the proposed multi-hop transformer. And also some experiments are provided  to illustrate whether the most attended object is the most important one.

Therefore, I am still standing on the previous justification for accepting the submitted manuscript.

---

> ### Author Response · Authors · 2020-11-20
> **Ablation and justification are added**
>
> We thank the reviewer for the valuable feedback and encouraging comments! We appreciate you for finding our idea “intuitive and interesting”, paper “written and presented well” with “thorough ablation” and “superior qualitative results”. We also thank the reviewer for the link to gradient-based visualization technique. We plan to incorporate it in the future. Below, we present our response in detail.
>
> **1. Regarding “one tracking baseline comparison is missing”:
> ===>**
>
> Thanks for the suggestion! We’ve added this tracking baseline in our updated manuscript. For every test video, we obtain the snitch track (based on which track's first object has snitch as its label) produced from our Hungarian algorithm, and project the center point of the bounding box of the last object in that track to the grid class. We also try with the majority vote, i.e., obtain the snitch track as the track who has the highest number of frames classified as snitch. We report the results of the majority vote method in Table 1 and 2, because it is a more robust method. The results of using the first frame are 31.8% Top-1, 40.2% Top-5, 2.7 L1 on CATER, and 28.2% Top-1, 36.3% Top-5, 2.8 L1 on CATER-h (details of this tracking baseline are available in the first paragraph of Appendix H.3).
>
> **2. Regarding “more ablation on the framework should be provided. How does the final performance benefit from using this transformer-based object detection model? Is the proposed framework only compatible with transformer-based object detection?”:
> ===>**
>
> The paper is updated as suggested. We’ve added additional ablations on the framework in Appendix A (Table 4) (e.g., ablating the tracking module). We’ve shown results of substituting Multi-hop Transformer with alternatives in Table 1 and Table 2. The biggest contribution of the work is the Multi-hop Transformer, not the object detector or tracker. The results in the paper suggest that Multi-hop Transformer adds significant improvements in performance (versus only ``"DETR + Hungarian" especially).
>
> The proposed framework is not just compatible with transformer-based object detectors. We use DETR because it is a state-of-the-art object detection model, and it is a transformer-based object detector unlike other methods (such as two stage detectors or the ones that require NMS). DETR is easy to integrate in an encoder-decoder architecture where the decoder can perform other downstream tasks (reasoning in our case). However, the proposed framework (and Multi-hop Transformer) sets no constraints on the actual type of object detector or the tracking technique.
>
> **3. Regarding “it is still a question whether the most attended object is the most important one in the final results or not”:
> ===>**
>
> We agree that in some cases, attention weights might not be interpretable and gradient visualization is another interesting and useful tool. To verify whether the most attended object is the most important one in the final results or not, we have conducted two new experiments: (1) we take our trained model, mask out the representation of the most attended object in the last hop by zeros, and then make predictions. This is to verify whether the most attended object in the last hop is important for the final Snitch Localization prediction task. Results of this are 32.5% Top-1, 60.9% Top-5, and 2.53 L1. (2) we take the trained model, mask out the representations of the most attended objects in all hops by zeros, and then make predictions. This is to verify how important are the most attended objects in all hops that are identified by our model. Results are 11.7% Top-1, 29.0% Top-5, and 3.60 L1. We believe the above results indicate that the attended objects identified by our model are important for the final results. This should attribute to our contrastive debiasing loss mentioned in Section 5 where we enforce consistency between attended objects and correct predictions, ensuring that the model understands why it is making a correct prediction.

---

### Official Review · AnonReviewer2 · 2020-10-29
**Official Blind Review #2**

**Rating:** 6
**Confidence:** 3

**Review:**

This paper proposes a multi-hop transformer method for the video-based object permanence task. The proposed method performs multi-hop reasoning via the encoder-decoder architecture of transformers over critical frames in the video. To mitigate the problem of lacking ground truth for the middle hops, the paper proposes some interesting training tricks. Overall, the paper is well organized and easy to follow.

Reasons to accept the paper:
1. The paper extends multi-hop reasoning techniques that are widely used in NLP domain to video domain, which may inspire other researchers working on other video-based tasks that require multi-hop reasoning.
2. The paper proposes a new benchmark dataset, which requires longer reasoning chains.
3. Experiments on the CATER dataset achieve state-of-the-art performance on the object permanence task.

Reasons to reject the paper:
1. The paper claims to address the problem of biased video reasoning, however, all the experiments in this paper are done on a synthetic dataset and it's not clear why such dataset can rule out the possibility of having biases.
2. The proposed method is somewhat dedicated to a proposed object permanence task, which may not be general enough to be extended to other video-based tasks.
3. The paper motivates the task using real-world examples, such as asking "which car was responsible for the accident". However, such question seems much harder to answer even with the proposed technique for object permanence task. Also, the paper only shows experimental results on synthetic dataset, and leaves experiments on real-world video datasets as future work.

---

> ### Author Response · Authors · 2020-11-20
> **Section 1 has been improved to address the concerns**
>
> Many thanks for appreciating our multi-hop reasoning techniques and recognizing their values! We appreciate you for finding our paper well organized and easy to follow. Please see our response in the following regarding the questions raised.
>
> **1. Regarding "all the experiments in this paper are done on a synthetic dataset and it's not clear why such dataset can rule out the possibility of having biases":
> ===>**
>
> We agree with the reviews the datasets contain biases. For example, we identified that the CATER dataset has a bias towards shorter reasoning chains and we address it with the CATER-h dataset. We’ve investigated the other potential dataset biases in Appendix G.6. Our finding highlights the staggering challenge of building a (synthetic or real) dataset that is completely unbiased. Thus, incorporating debiasing techniques into a model, like what we did, might be a better cure.
>
> For expanding our result to real dataset, Hopper needs tracking and object detection. Recent work on tracking using DeepSORT (Wojke et al. (2017)) as well as consistent improvements in object detection has shown that this is possible. Our contribution, Hopper uses these building blocks to provide a multi-hop reasoning solution. Moreover, a less biased real-world dataset for general video recognition and long-term reasoning to date is not available, to the best of our knowledge. The CATER dataset was recently proposed in ICLR 2020,  and is an excellent benchmark given that the task is far from being addressed.
>
> Based on your concerns, we’ve changed the sentence in the manuscript to say "we propose Hopper to address *debiased* video reasoning", since what our method can do is debiasing (with the contrastive debiasing loss).
>
>
> **2. Regarding "the proposed method is somewhat dedicated to a proposed object permanence task.":
> ===>**
>
> We believe that explicitly addressing the object permanence task is fundamental to reasoning in videos (Bottou, 2014). It has also been shown as one of the important steps in child development (Bremner et al., 2015). Many applications as described in Section 1, like “who hit the football, resulting in a goal”, “what items eventually a shopper walked out with” etc. rely on the ability to reason about objects and associated entities as they move in videos.
>
> In addition, the proposed methods are not dedicated to the object permanence task. The modular Hopper framework that we proposed, is applicable to any video understanding task (e.g., action classification). Hopper first obtains frame and object representations, then computes object tracks to have more consistent object representations. Finally, Hopper uses a reasoning module to perform reasoning of a video. This Hopper framework is general. On the other hand, the proposed Multi-hop Transformer, the reasoning module, is applicable to any task that involves sequences and requires multi-step reasoning over such sequences.
>
> **3. Regarding "the paper motivates the task using real-world examples, such as asking 'which car was responsible for the accident'. However, such question seems much harder to answer even with the proposed technique for object permanence task.":
> ===>**
>
> We agree with the reviewer that the question "which car was responsible for the accident" may require additional information or other techniques (like causality) to fully address it in all cases. Nevertheless, learning object permanence is crucial for the model to comprehend that it is a single object that appears and disappears even when it is partially or fully occluded.
>
> We've improved the Introduction section and removed the sentence on associating our task with the question, "which car was responsible for the accident?". However, we believe object permanence can be essential in understanding videos of driving, e.g., to infer "is there a car next to me in the right lane?". Because if a car is there in the right lane, the car may appear and disappear, and humans usually know that its the same car.

---

### Author Response · Authors · 2020-11-20
**Submission Update**

We thank all reviewers for their constructive comments. Our contributions, as identified by the reviewers include a general and modular Hopper framework (R1,4), Multi-hop Transformer as a reasoning module (R1,2,3,4), and the release of the new CATER-h dataset that requires long reasoning chains (R1,2,4). We have conducted comprehensive experiments (R1,2,3), ablations (R1,3,4), visualizations (R1,3) to showcase the benefits of our approach, in comparison to state-of-the-art.

In accordance with reviewer feedback, we have made the following notable changes to the revised manuscript:
1. We’ve significantly improved the clarity of the paper. Modifications have been done to Section 1 (Introduction), Section 3.3 (for motivation of Tracking), Section 4 (Multi-hop Transformer), Section 5 (added a “Summary & discussion” subsection), Appendix A, B, C.1, G.6, etc.
2. We’ve added results of ablative experiments (Appendix A) to address concerns from reviewers (R1,3,4).
3. We’ve conducted new analysis on the model (Appendix B, C.1) and dataset (Appendix G.6) to incorporate the feedback from reviewers (R1,2,3,4).

Thanks again for all of the questions which have helped in improving our paper!

---

### Decision · Program_Chairs · 2021-01-07
**Final Decision**

**Decision:**

Accept (Poster)

**Comment:**

This paper was reviewed by four experts in the field. Based on the reviewers' feedback, the decision is to recommend the paper for acceptance to ICLR 2021. The reviewers did raise some valuable concerns that should be addressed in the final camera-ready version of the paper. The authors are encouraged to make the necessary changes.  It is also very important to think about how to extend this framework to the more challenging CLEVERER dataset (http://clevrer.csail.mit.edu/).